# Integrated experimental-computational analysis of a HepaRG liver-islet microphysiological system for human-centric diabetes research

**Belén Casas**[1,2], **Liisa Vilén**[1], **Sophie Bauer**[3], **Kajsa P. Kanebratt**[1], **Charlotte Wennberg Huldt**[4], **Lisa Magnusson**[4], **Uwe Marx**[3], **Tommy B. Andersson**[1], **Peter Gennemark**[1,2‡], **Gunnar Cedersund**[2,5‡]*

**1** Drug Metabolism and Pharmacokinetics, Research and Early Development, Cardiovascular, Renal and Metabolism (CVRM), BioPharmaceuticals R&D, AstraZeneca, Gothenburg, Sweden, **2** Department of Biomedical Engineering, Linköping University, Linköping, Sweden, **3** TissUse GmbH, Berlin, Germany, **4** Bioscience, Research and Early Development, Cardiovascular, Renal and Metabolism (CVRM), BioPharmaceuticals R&D, AstraZeneca, Gothenburg, Sweden, **5** Center for Medical Image Science and Visualization (CMIV), Linköping University, Linköping, Sweden

‡ These authors are shared senior authors on this work.
* gunnar.cedersund@liu.se

## Abstract

Microphysiological systems (MPS) are powerful tools for emulating human physiology and replicating disease progression *in vitro*. MPS could be better predictors of human outcome than current animal models, but mechanistic interpretation and *in vivo* extrapolation of the experimental results remain significant challenges. Here, we address these challenges using an integrated experimental-computational approach. This approach allows for *in silico* representation and predictions of glucose metabolism in a previously reported MPS with two organ compartments (liver and pancreas) connected in a closed loop with circulating medium. We developed a computational model describing glucose metabolism over 15 days of culture in the MPS. The model was calibrated on an experiment-specific basis using data from seven experiments, where HepaRG single-liver or liver-islet cultures were exposed to both normal and hyperglycemic conditions resembling high blood glucose levels in diabetes. The calibrated models reproduced the fast (i.e. hourly) variations in glucose and insulin observed in the MPS experiments, as well as the long-term (i.e. over weeks) decline in both glucose tolerance and insulin secretion. We also investigated the behaviour of the system under hypoglycemia by simulating this condition *in silico*, and the model could correctly predict the glucose and insulin responses measured in new MPS experiments. Last, we used the computational model to translate the experimental results to humans, showing good agreement with published data of the glucose response to a meal in healthy subjects. The integrated experimental-computational framework opens new avenues for future investigations toward disease mechanisms and the development of new therapies for metabolic disorders.

**Data Availability Statement:** All data supporting the results in this study are available in the main article and the Supplementary Information. Codes to reproduce all the figures are available at https://github.com/belencasasgarcia/Liver-islet-MPS.git.

**Funding:** The authors acknowledge funding from the Swedish Research Council [https://www.vr.se/], grant numbers: 2018-05418 and 2018-03319 (GC), CENIIT [http://ceniit.lith.liu.se//], grant number: 15.09 (GC), the Swedish Foundation for Strategic Research [https://strategiska.se/en/], grant number: ITM17-0245 (GC), the SciLifeLab/KAW [https://www.scilifelab.se/] National COVID-19 Research Program, financed by the Knut and Alice Wallenberg Foundation, grant number: 2020.0182 (GC), and the H2020 project PRECISE4Q [https://precise4q.eu/], grant number: 777107 (GC). Additional support for GC came from The Swedish Fund for Research without Animal Experiments [https://forskautandjurforsok.se/], grant number: F2019-0010, and ELLIIT [https://elliit.se/], grant number: 2020-A12. The funders had no role in study design, data collection and analysis, decision to publish, or preparation of the manuscript.

## Author summary

Microphysiological systems (MPS) are powerful tools to unravel biological knowledge underlying disease. MPS provide a physiologically relevant, human-based *in vitro* setting, which can potentially yield better translatability to humans than current animal models and traditional cell cultures. However, mechanistic interpretation and extrapolation of the experimental results to human outcome remain significant challenges. In this study, we confront these challenges using an integrated experimental-computational approach. We present a computational model describing glucose metabolism in a previously reported MPS integrating liver and pancreas. This MPS supports a homeostatic feedback loop between HepaRG/HHSteC spheroids and pancreatic islets, and allows for detailed investigations of mechanisms underlying type 2 diabetes in humans. We show that the computational model captures the complex dynamics of glucose-insulin regulation observed in the system, and can provide mechanistic insight into disease progression features, such as insulin resistance and β-cell dynamics. Furthermore, the computational model can explain key differences in temporal dynamics between MPS and human responses, and thus provides a tool for translating experimental insights into human outcome. The integrated experimental-computational framework opens new avenues for future investigations toward disease mechanisms and the development of new therapies for metabolic disorders.

## 1 Introduction

Type 2 diabetes mellitus (T2DM) is a complex multifactorial disease characterized by impaired glucose homeostasis. In healthy individuals, plasma glucose levels are maintained within a narrow range (3–9 mM) [1] via a negative feedback between glucose and insulin. Insulin is secreted in response to elevated glucose levels, increasing glucose uptake in target tissues (adipose, muscle and liver) to restore normoglycemia [2]. In early stages of the development into T2DM, target tissues become insulin resistant and require higher insulin concentrations to maintain normal glucose levels [3]. Initially, β cells compensate for insulin resistance through upregulation of insulin secretion (β-cell adaptation), but over time they may be unable to meet the increased insulin demand and overt T2DM manifests [4]. While these general steps in the disease etiology are well established, more detailed knowledge of the interplay between insulin resistance, pancreatic β-cell adaptation, and the progression of T2DM is still missing.

Currently, research on the pathogenesis and potential therapeutic agents in T2DM is primarily based on *in vivo* animal models [5]. However, the translatability of these animal-based studies to human outcome is often limited. One fundamental obstacle for this translation is the naturally existing phylogenetic difference between the animals typically used in preclinical testing and humans. Some mouse strains commonly used in T2DM research, such as C57BL/6J, develop insulin resistance on a high-fat diet [6] and a subsequent upregulation of insulin secretion [7]. However, unlike in humans, this increase in insulin secretion is not followed by β-cell failure, which is one of the hallmark features of human T2DM [8]. Lean rodent models have also been applied in T2DM research [9], but the development of impaired glucose homeostasis in these models seems to be a consequence of aberrant β-cell mass [10] and/or β-cell function [11] rather than insulin resistance. While these models can still be valuable for mechanistic and mode of action studies, the majority of drug candidates that show promise in preclinical animal studies ultimately fail to result in functional and safe drugs in humans [12–14]

Because of aforementioned limitations in using animal studies, there is a critical need for novel preclinical models that can better represent human physiology and predict *in vivo* outcomes. This need has fueled the development of microphysiological systems (MPS), which are microscale devices capable of replicating human physiology *in vitro*. By integrating cultures of human organ-specific cells in a microfluidic platform, these *in vitro* systems aim to recreate key microenvironmental aspects of *in vivo* tissues (flow, multicellular architectures, and tissue-tissue interfaces), thereby being more physiologically relevant than standard cell cultures [15–17].

We have previously presented a two-organ MPS integrating liver and pancreas, which offers an advantage over single-organ MPS for studying glucose homeostasis [18]. Recent advances in MPS technology have led to the development of single organ-MPS for both liver [19–23] and pancreas [24], which are two major organs involved in the maintenance of glucose homeostasis. However, single organ-MPS have limited relevance for studying metabolic diseases like T2DM, as the underlying pathophysiology involves disruption in the homeostatic cross-talk between several organs. Therefore, multi-organ platforms capable of capturing interactions between two or more organs are best suited for investigating these diseases *in vitro*. Our previously developed HepaRG liver-islet MPS supports a homeostatic feedback loop between co-cultured HepaRG/HHSteC spheroids and pancreatic islets [18]. This MPS allows detailed investigations of mechanisms underlying T2DM through enabling changes in both the operating and co-culture conditions in a controlled and systematic manner. For instance, one could perform changes in the glycemic levels or the composition of the co-culture medium, as well as in the number and metabolic functions of the co-cultured cells. Moreover, unlike other *in vivo* and *in vitro* models, this experimental setup offers great flexibility to study interactions between specific subsets of organs. Therefore, assays based on this system could become superior to animal experiments for studying disease progression and drug metabolism.

However, to improve the applicability of our HepaRG liver-islet MPS, two major challenges should be addressed. First, there is still limited mechanistic understanding of the physiological processes in the MPS. Elucidating these mechanisms using a purely experimental approach would be challenging, as the biological processes underlying glucose-insulin regulation in the MPS are complex, non-linear, and involve numerous feedback loops. Because of these complexities, relying on qualitative analysis and statistics of the experimental data may often lead to incorrect conclusions [25,26]. Second, the experimental findings from the system cannot be directly extrapolated to *in vivo*, human outcome. Although existing strategies for on-platform scaling could be applied to achieve *in vitro* responses that better mimic those observed *in vivo*, these have proven insufficient to ultimately establish the translation to humans [27]. These two challenges could be confronted using computational modelling. More specifically, computational modelling provides a framework to quantitatively represent the system including its nonlinearities and feedback loops, integrate and interpret the experimental measurements, infer physiological variables that cannot be directly measured *in vitro*, and enable *in vitro* to *in vivo* translation.

While several studies have shown the added value of combining computational models with multiorgan MPS for data interpretation, these have mainly focused on pharmacokinetic (PK) [28–31] and in some cases pharmacokinetic/pharmacodynamic (PKPD) strategies [32,33], rather than providing mechanistic understanding of the underlying physiology. Some efforts have been done to integrate MPS with more descriptive models, often referred to as quantitative systems pharmacology (QSP) models [27,34]. These models generally incorporate physiological parameters describing the MPS operating conditions, such as organoid sizes and flow rates, and more mechanistic knowledge about the physiology of the system. However, to date, the number of studies exploring this approach is still limited [35–38], especially for studying glucose metabolism. A few studies have also focused on computational strategies to extrapolate the *in vitro*

responses to human proportions. Typically, these strategies account for constraints in the *in vitro* setting that limit the capability of the MPS to reproduce human-like responses. These constraints generally include sizes of the organoids and the co-culture media, mismatches in media-to-tissue ratio and the fact that some organs and functions are missing in the MPS [39,40]. In a recent study, Lee et al. [41] presented a computational model for a pancreas-muscle MPS to study glucose metabolism, and added a liver compartment *in silico* to improve the physiological relevance of glucose and insulin dynamics in the system. However, the model was constructed using experimental data from individual cultures of myoblasts and pancreatic cells from rodents, and did not incorporate measurements from interconnected co-cultures that could reflect organ cross-talk. Despite the crucial role of the liver-pancreas cross-talk in maintaining glucose homeostasis, the combination of a mechanistic computational model and an MPS emulating the interaction between these organs has not been investigated yet.

In this study, we propose combining our HepaRG liver-islet MPS with a computational model to augment *in vitro* investigations of human glucose homeostasis in healthy and hyperglycemic conditions mimicking high blood glucose levels in T2DM. Our aim is to use the model to integrate and quantitatively analyze the experimental data to improve their mechanistic interpretation, generate model predictions and, ultimately, extrapolate the results from *in vitro* to *in vivo*. The possibility to generate human-relevant predictions from *in vitro* experiments at a relatively low cost could reduce the need for animal models of T2DM in the future.

## 2 Materials and methods

### 2.1 *In vitro* experiments

To construct, calibrate and evaluate the computational model we used data from seven independent *in vitro* MPS experiments (S1 Table). Each experiment corresponds to a different donor of pancreatic islets and involves measurements on several chips, each of them including two platform replicates (Fig 1A and 1B). Across all experiments, the number of platform replicates used in the experiment was 5 ± 2 (range 4–10), Two of the experiments (experiments 1 and 2) have already been published in [18]. In the following section, we describe the materials and methods for the five experiments performed for this study.

**2.1.1 Multi-organ chip platform.** To co-culture HepaRG/HHSteC spheroids and pancreatic islet microtissues (from now on referred to as HepaRG/HHSteC spheroids and pancreatic islets, respectively), we used the Chip2 from TissUse which allows for simultaneous culture of two organ models in spatially separated, but interconnected culture compartments (Fig 1A and 1B). Details on the design and fabrication process of the Chip2 are described in prior publications [42,43]. The culture compartments are connected by a microfluidic channel, with an on-chip micropump driving a continuous pulsatile flow that supports long-term perfusion of the chip-cultured three-dimensional (3D) cell constructs. Both culture compartments contain 300 $\mu$L of culture medium and the microfluidic channels hold additional 5 $\mu$L of medium. The average volumetric flow rate between the compartments was set to 4.94 $\mu$L/min, resulting in an approximate medium turnover time of 2 h.

**2.1.2 Pre-culture of HepaRG/HHSteC spheroids and pancreatic islets.** Human, HepaRG/HHSteC spheroids were formed in 384-well spheroid microplates (3830, Corning) by combining differentiated precultured HepaRG cells (Lot HPR116189, HPR116239, HPR116246, HPR116NS080003 or HPR116222; Biopredic International; France) and passaged primary human hepatic stellate cells (HHSteC) (Lot PFP; passage 4–6, BioIVT; USA) at a rat io of 24:1 as previously described [18]. The HepaRG cells are terminally differentiated hepatic cells derived from a human liver progenitor cell line, which retain a similar expression profile compared to primary human hepatocytes of genes relevant for the glucose metabolism and

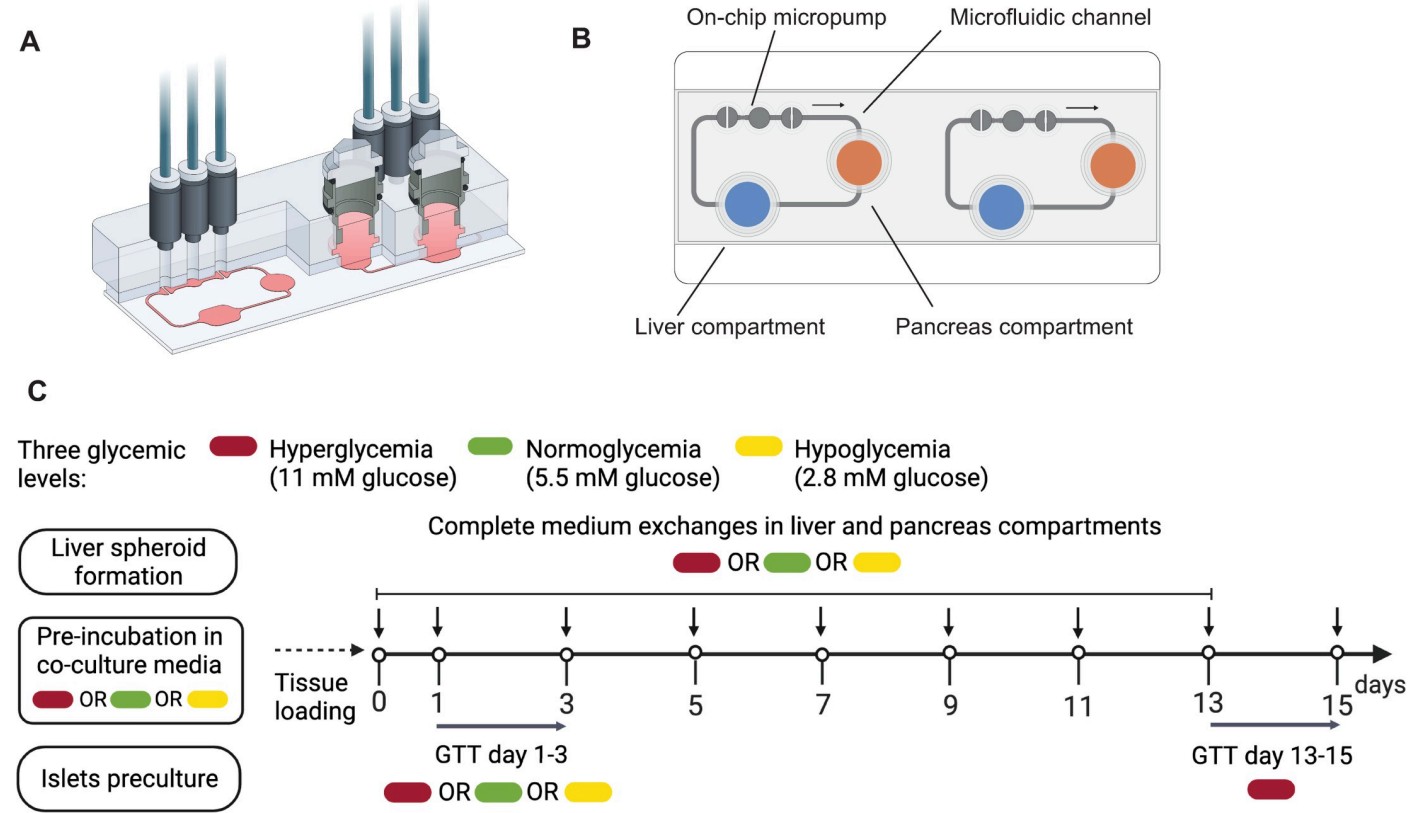

**Fig 1. *In vitro* experiments.** (A) A 3D view of the Chip2 (copyright by TissUse GmbH). (B) Illustration of the Chip2 from underneath, including liver (blue) and pancreas (orange) compartments, the microfluidic channel, and the on-chip micropump. The arrow indicates the direction of flow between culture compartments. The Chip2 comprises two replicate platforms. (C) Generic study design used in the experiments performed in the study. In five independent MPS experiments (N = 5), co-cultures or single organ-cultures were exposed to either hyperglycemia (11 mM glucose, N = 5, red), normoglycemia (5.5 mM glucose, N = 3, green) or hypoglycemia (2.8 mM glucose, N = 1, yellow). Medium exchanges occurred every 48 h (vertical black arrows) and glucose tolerance tests (GTTs) were performed on indicated days (horizontal gray arrows). GTT d13-15 was initiated by adding co-culture medium with 11 mM glucose into each of the culture compartments. In GTT d1-3, the glucose concentration in the co-culture medium was set to the glycemic level corresponding to each regime (11 mM, 5.5 mM and 2.8 mM for co-cultures exposed to hyperglycemia, normoglycemia and hypoglycemia, respectively). Samples of the medium were taken 0, 8, 24 and 48 h after the start of the GTT. Figure created with BioRender.

remain responsive to insulin [18,44]. HepaRG pre-cultures and HepaRG/HHSteC spheroids were maintained in Williams' medium E supplemented with 10% FBS, 2 mM L-glutamine, 50 μM hydrocortisone hemisuccinate, 50 μg/ml gentamycin sulfate and 0.25 μg/ml amphotericin B. Glucose and insulin concentrations in the HepaRG maintenance medium were set according to the glycemic level used in the MPS culture. For cultures in hyperglycemia, normoglycemia or hypoglycemia, the pre-culture medium contained 11 mM glucose and 860 nM insulin, 5.5 mM glucose and 1 nM insulin, or 2.8 mM glucose and 0.1 nM insulin, respectively. The medium used for the pre-cultured HepaRG cells was additionally supplemented with 2% DMSO, while this was omitted in the medium for spheroid formation in order to avoid harmful effects on the human hepatic stellate cells. Human pancreatic islets were purchased from InSphero (MT-04-002-0; Switzerland) and maintained in 3D InSight Human Islet Maintenance Medium (CS-07-005-02; InSphero) until the MPS culture. The pancreatic islets used in this study were reaggregated from dispersed human pancreatic islets retaining the composition of $\alpha$, $\beta$ and $\delta$ cells representative of normal human pancreatic islets, and did not include exocrine cells.

The pancreatic islet donors included five men, with an age of 54 ± 5 years (range 45–57 years), body mass index (BMI) 28 ± 2 kg/m$^2$ (range 27–30 kg/m$^2$), hemoglobin A1c (HbA1c)

5.6 ± 0.3% (range 5.1–5.8%) with no known history of diabetes. All cell cultures were maintained at 37°C and 5% $CO_2$.

**2.1.3 MPS cultures.**   Before transferring HepaRG/HHSteC spheroids and pancreatic islets into the Chip2, both were washed twice with 0.1% BSA in 1xPBS and pre-incubated in insulin-free HepaRG maintenance medium for 2 hours. In our previous study, we showed that the HepaRG/HHSteC spheroids were sensitive to insulin following the pre-culture period and prior to the beginning of the co-culture, by quantifying AKT phosphorylation [18]. To setup the co-culture, 40 HepaRG/HHSteC spheroids and 10 pancreatic islets were placed into the liver and pancreas compartments, respectively. In comparison to their respective human counterparts, this corresponds to a downscaling factor in the order of 100,000 in both organs [45,46]. In single-liver cultures, 40 HepaRG/HHSteC spheroids were added into the liver compartment while keeping the pancreas compartment empty. Both the co-cultures and single-organ cultures were maintained in insulin-free HepaRG medium (referred as co-culture medium from here on) with glucose concentration of 11 mM, 5.5 mM or 2.8 mM depending on the glycemic regime (Fig 1C). The co-culture culture medium with respective glucose level was changed first after 24 hours and then after every 48 hours over the culture period of 15 days. In each individual experiment, all glycemic levels were studied in 4–10 replicates.

**2.1.4 *In vitro* glucose tolerance test.**   Regulation of glucose homeostasis as a result of organ cross-talk was assessed by *in vitro* glucose tolerance tests (GTTs) [18]. On day 13, 315 μL of co-culture medium with 11 mM glucose was added into each of the two culture compartments and samples of 15 μL were collected after 0, 8, 24 and 48 hours. The samples collected from both compartments were pooled together for glucose and insulin analysis, resulting in a maximal volume decrease of 10% over the entire GTT. Additionally, a similar sampling scheme was performed on day 1 but keeping the respective glycemic level as indicated for each regime.

**2.1.5 Glucose and insulin analysis.**   Glucose concentrations were determined either using the GLU 142 kit (Diaglobal, Berlin, Germany) as described previously [18] or using Glucose Liquid Reagent (1070–400, Stanbio) with minor modifications to the manufacturer's instructions. Briefly, 5 μL of culture supernatant was mixed with 95 μL of pre-heated assay reagent and after 5 min incubation at 37°C the absorbance was measured at 520 nm. Insulin concentrations were measured using Insulin ELISA (10-1113-01, Mercodia) following the manufacturer's instructions.

**2.1.6 Glucose-stimulated insulin secretion.**   After the MPS culture, pancreatic islets were transferred from the chips into a GravityTRAP plate (InSphero) to analyse their glucose-stimulated insulin secretion (GSIS) as earlier described [18]. In brief, pancreatic islets were equilibrated in low glucose buffer (2.8 mM) for 2 h followed by sequential 2 h incubations first in low glucose buffer and then in high glucose buffer (16.8 mM).

## 2.2 Computational model of the HepaRG liver-islet MPS

We developed a computational model describing glucose metabolism in the HepaRG liver-islet MPS. The model is based on data from seven independent experiments corresponding to seven different donors of pancreatic islets (N = 7). The model, outlined in Fig 2, describes key physiological processes underlying glucose regulation on a short-term (meal response) basis and the long-term changes in insulin resistance and β-cell adaptation associated with impaired glucose homeostasis. We included two model compartments, each of them representing a specific organoid (liver or pancreas) and its corresponding co-culture medium. The compartments are connected in a closed loop, and the medium circulates as specified by a flow rate parameter.

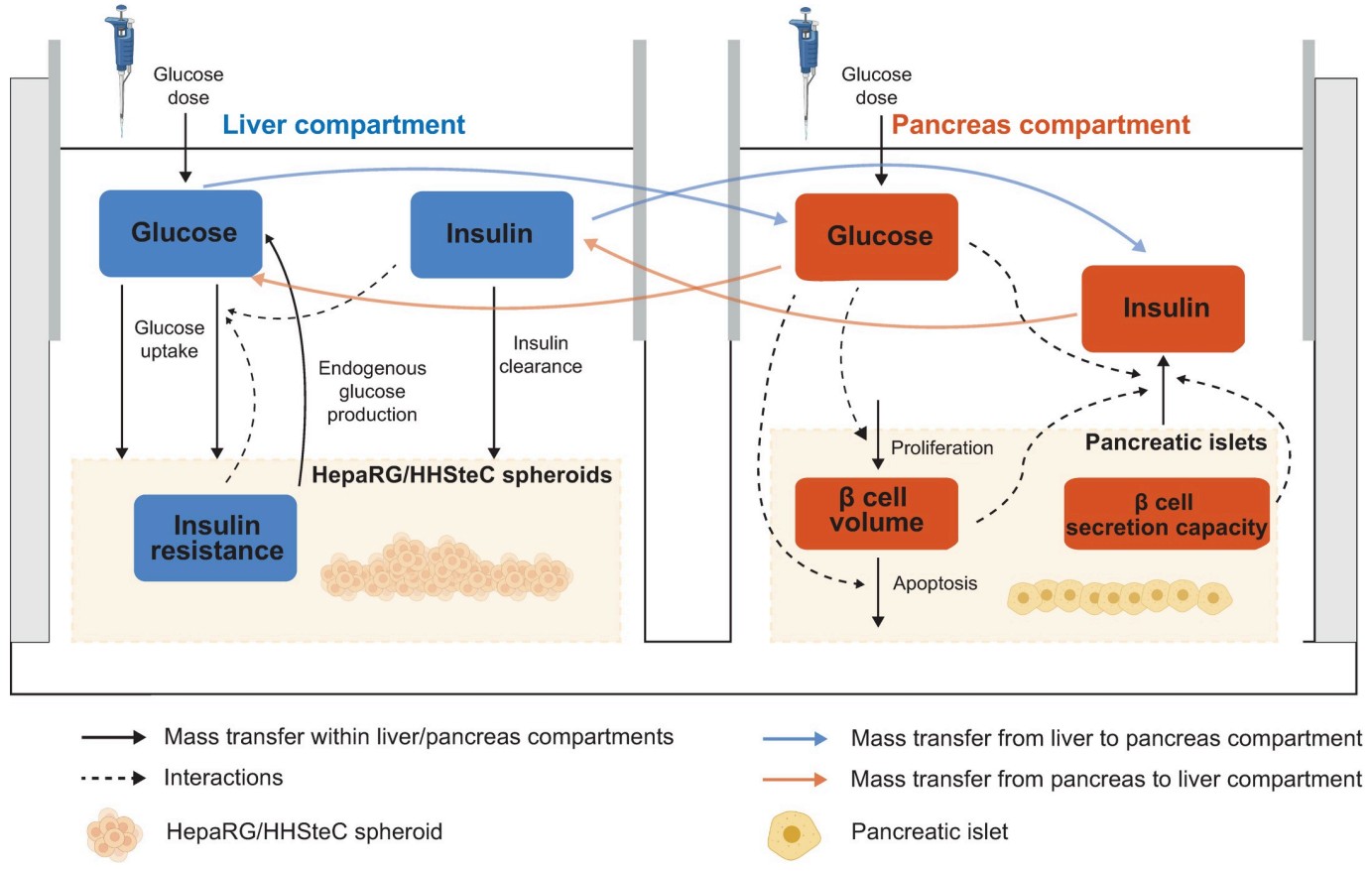

**Fig 2. Graphical illustration of the computational model of the HepaRG liver-islet MPS.** The physiological variables described by the model are displayed in blue and red text boxes. The solid arrows represent changes in these variables, mainly because of metabolic fluxes within each organoid compartment (black arrows) or between compartments (blue and orange arrows). Interactions between the variables are represented as dashed arrows.

The model is based on the long-term glucose, insulin and β-cell mass dynamics proposed by Topp et al. [47]. Here, we have modified this model to: 1) encompass two model components with different time scales: a fast (hours) component for glucose and insulin dynamics between media exchanges, and a slow (weeks) component describing the development of hepatic insulin resistance and *β*-cell adaptation, 2) explicitly establish an interaction between the fast and slow model components, allowing short-term dynamics of physiological variables to impact long-term progression of the disease (e.g. effects of daily glucose levels on insulin resistance and *β*-cell volume dynamics), 3) support scaling to humans by specifying organ sizes and operating conditions in the MPS (i.e. flow rate between culture compartments and co-culture media volumes) and 4) allow inhibition in β-cell insulin secretion over time.

The model was formulated using a system of mass-balanced, non-linear ordinary differential equations (ODEs), and well-mixed conditions were assumed in each compartment. A complete description of the model equations including definitions of the metabolic fluxes in the system is provided in S1 Appendix.

**2.2.1 Modelling short-term glucose homeostasis in the co-culture.** The short-term glucose homeostasis was modelled in the following way. First, we based our description of glucose on the glucose equation in the model of Topp et al. [47]. This equation derives from a simplification of the minimal glucose model [48] to represent daily average glucose [49]. Glucose content in the liver compartment culture medium is controlled by glucose dosing to the system,

endogenous glucose production and glucose uptake by the HepaRG/HHSteC spheroids, as well as glucose inflow from and outflow to the pancreas compartment:

$$\frac{dNG_{m,liver}(t)}{dt} = G_d(t) + V_{HepaRG,spheroids} \cdot EGP(t) - V_{HepaRG,spheroids} \cdot \left( E_{G0} + S_I(t) \cdot \frac{NI_{m,liver}(t)}{V_{m,liver}} \right) \cdot$$

$$\frac{NG_{m,liver}(t)}{V_{m,liver}} + Q \cdot \frac{NG_{m,pancreas}(t)}{V_{m,pancreas}} - Q \cdot \frac{NG_{m,liver}(t)}{V_{m,liver}} \ (mmol/h) \tag{1}$$

where the symbols in the equation are defined as follows. $NG_{m,liver}(t)$ and $NG_{m,pancreas}(t)$ are the number of glucose molecules (mmol) in the culture media corresponding to the liver and pancreas compartments, respectively, and $NI_{m,liver}(t)$ is the number of insulin molecules in the liver compartment's co-culture medium (mIU). The glucose input rate $G_d(t)$ (mmol/h) represents glucose variations due to media exchanges, and $EGP(t)$ describes endogenous glucose production in the HepaRG/HHSteC spheroids (mmol/L/h). After analysing the experimental data, $EGP(t)$ was concluded to be negligible based on the observed decline in glucose levels below normoglycemia (5.5 mM) in the system. Therefore, in practice, $EGP(t)$ was set to zero. Glucose uptake by the HepaRG/HHSteC spheroids is largely dependent on the insulin-independent glucose disposal rate (denoted $E_{G0}$ (1/h)), but is also enhanced by the action of insulin. This enhancement accounts for an increased glucose influx through the GLUT2 hepatic transporter as a result of the reduction in intracellular glucose via insulin-induced metabolic pathways (e.g. glycogen synthesis and *de novo* lipogenesis) [3,50]. The variable $S_I(t)$ (L/mIU/h) denotes the insulin sensitivity of the HepaRG/HHSteC spheroids. The parameters representing operating conditions include the flow rate between culture compartments (denoted $Q$ (L/h)), the total volume of HepaRG cells in the HepaRG/HHSteC spheroids ($V_{HepaRG,spheroids}$ ($L$)) and the volume of co-culture medium in the liver and pancreas compartments ($V_{m,liver}$ and $V_{m,pancreas}$ (L), respectively), as previously described by Lee et al. [41].

Second, the release of insulin from β-cells in the pancreatic islets was modelled as a sigmoidal function of glucose concentration [49,51,52]:

$$\frac{dNI_{m,pancreas}(t)}{dt} = V_{\beta,islets}(t) \cdot \sigma(t) \cdot \frac{\left( \frac{NG_{m,pancreas}(t)}{V_{m,pancreas}} \right)^2}{EC50_I^2 + \left( \frac{NG_{m,pancreas}(t)}{V_{m,pancreas}} \right)^2} + Q \frac{NI_{m,liver}(t)}{V_{m,liver}}$$

$$- Q \frac{NI_{m,pancreas}(t)}{V_{m,pancreas}} \ (mIU/h) \tag{2}$$

where $NI_{m,pancreas}(t)$ and $NI_{m,liver}(t)$ are the number of insulin molecules (mIU) in the pancreas and the liver compartment, respectively. Insulin secretion depends on the volume of β cells in the pancreatic islets (denoted $V_{\beta,islets}(t)$ (L)), the insulin secretion capacity per unit volume of β cells (denoted $\sigma(t)$ (mIU/L/h)), and the glucose concentration resulting in half-of-maximum response to insulin (denoted $EC50_I$ (mmol/L)). We account for a decrease in the insulin secretion capacity of the β cells over time, as given by:

$$\sigma(t) = \sigma_{max} \cdot \left( 1 - \frac{t^2}{\alpha + t^2} \right) (mIU/L/h) \tag{3}$$

Here, we assume that this decrease follows a sigmoidal dependence on time, determined by the parameter $\alpha$ (h$^2$). The parameter $\sigma_{max}$ (mIU/L/h) represents the maximal insulin secretion rate of the β cells (i.e. at the beginning of the co-culture). The parameter $\alpha$ is estimated on an experiment-specific basis through optimization against the experimental measurements

(Section 2.3, S3 Table). For large values of this parameter, the decrease in insulin secretion capacity over time would be negligible.

**2.2.2 Modelling hepatic insulin resistance and β-cell dynamics.** The dynamics of hepatic insulin sensitivity $S_I(t)$ were modelled under the assumption that insulin responsiveness of the co-cultured HepaRG/HHSteC spheroids decreases because of sustained exposure to hyperglycemia:

$$S_I(t) = S_{I0} \cdot \left( 1 - \frac{I_{max,Si} \cdot G_{int}(t)}{EC50_{Si} + G_{int}(t)} \right) \ (L/mIU/h) \tag{4}$$

$$\frac{dG_{int}(t)}{dt} = \begin{cases} \dfrac{NG_{m,liver}(t)}{V_{m,liver}} - G_{normo} & \dfrac{NG_{m,liver}(t)}{V_{m,liver}} - G_{normo} \geq 0 \\[3ex] 0 & \dfrac{NG_{m,liver}(t)}{V_{m,liver}} - G_{normo} < 0 \end{cases} \left( \frac{mmol}{L} \right) \tag{5}$$

where $S_{I0}$ (L/mIU/h) is the insulin sensitivity at the start of the co-culture. $S_I(t)$ decreases progressively as the HepaRG/HHSteC spheroids are exposed to glucose levels above the normo-glycemic range in the co-culture medium, which we refer to as excess glucose ($\frac{NG_{m,liver}(t)}{V_{m,liver}} - G_{normo} \geq 0$). $G_{int}(t)$ represents the integral of excess glucose over time. The proposed model captures the hypothesis postulated by Bauer et al. [18] that even short hyperglycemic periods (<24 h) could induce insulin resistance in the co-cultured HepaRG/HHSteC spheroids, resulting in an increase in glucose levels over time as observed in our system. This is also consistent with the results from Davidson et al. [53], which reported the development of insulin resistance in primary human hepatocytes after six days of exposure to a hyperglycemic culture medium containing 25 mM glucose. In our model, the decrease in $S_I(t)$ was represented by a sigmoidal function with maximal fractional reduction $I_{max,Si}$, and with half of the maximal fractional reduction occurring at $EC50_{Si}$ (mmol h/L).

In the computational model, insulin secretion depends on both the total volume of β cells in the pancreatic islets and their individual secretion capacity. The β cells adapt to the long-term (slow) changes in glucose concentration by regulating their rates of replication and apoptosis, as previously described by Topp et al. [47]. This adaptation changes the number of β cells, with an associated change in total β-cell volume ($V_{\beta,islets}(t)$) given by:

$$\frac{dV_{\beta,islets}(t)}{dt} = (Replication - Apoptosis) \cdot V_{\beta,islets}(t) \ (L/h) \tag{6}$$

$$Replication = k_v \cdot \left( r_{1,r} G_{slow,pancreas}(t) - r_{2,r} G_{slow,pancreas}(t)^2 \right) \tag{7}$$

$$Apoptosis = k_v \cdot \left( d_0 - r_{1,a} G_{slow,pancreas}(t) + r_{2,a} G_{slow,pancreas}(t)^2 \right) \tag{8}$$

where the rates of replication and apoptosis are modelled as nonlinear functions of glucose concentration in the medium, on the basis of previous *in vitro* studies [54–57]. The parameter $d_0$ is the death rate at zero glucose (h$^{-1}$) and $r_{1,r}$, $r_{2,a}$ (L/mmol/h), $r_{2,r}$, $r_{1,a}$ (L$^2$/mmol$^2$/h) are parameters that determine the dependence of the replication and apoptosis rates on glucose. The parameter $k_v$ was introduced to account for potential differences in behaviour between human pancreatic islets in our *in vitro* system and rodent islets in the model of Topp et al.

[47]. This parameter is estimated on an experiment-specific basis through optimization against the experimental measurements (Section 2.3, S3 Table)

We have modified the original insulin secretion model [47] by introducing the variable $G_{slow,pancreas}(t)$, which represents the long-term average glucose concentration in the co-culture medium, as given by:

$$\frac{dG_{slow,pancreas}(t)}{dt} = \frac{G_{pancreas}(t) - G_{slow,pancreas}(t)}{\tau_{slow}} \ (mmol/L/h) \tag{9}$$

where $G_{pancreas}(t)$ is calculated from the number of glucose molecules in the co-culture medium corresponding to the islets compartment $NG_{m,pancreas}(t)$ (mmol) and $V_{m,pancreas}$ $(G_{pancreas}(t) = {}^{NG_{m,pancreas}(t)}/_{V_{m,pancreas}})$ and $\tau_{slow}$ (h) is a time constant that determines the averaging of $G_{pancreas}(t)$ over time. Previous *in vivo* and *in vitro* studies on rodent pancreatic islets have demonstrated changes in β-cell mass and proliferation via glucose stimulation on a time scale of days [54,57–60]. Therefore, the value of $\tau_{slow}$ was chosen so that $G_{slow,pancreas}(t)$, represents daily average glucose in the co-culture medium.

The equation describing the dynamics of β-cell volume (Eq 6) can then be rewritten as follows:

$$\frac{dV_{\beta,islets}(t)}{dt} = k_v\Big(-d_0 + r_1 G_{slow,pancreas}(t) - r_2 G_{slow,pancreas}(t)^2\Big) \cdot V_{\beta,islets}(t) \ (L/h) \tag{10}$$

where $r_1 = r_{1,r} + r_{1,a}$ (L/mmol/h) and $r_2 = r_{2,r} + r_{2,a}$ (L$^2$/mmol$^2$/h). The formulation for the rate of change of β-cell number ($k_v(-d_0 + r_1 G_{slow,pancreas}(t) - r_2 G_{slow,pancreas}(t)^2$) captures the hypothesis that a small increase in glucose from normoglycemia (i.e. mild hyperglycemia) leads to an increase in total β-cell volume in order to restore glucose homeostasis, while a higher glucose concentration drives total β-cell volume down instead [47,49]. Based on the study from Topp et al. [47], the values of $r_1$ and $r_2$ were chosen to achieve two steady state solutions at glucose concentrations corresponding to 5.55 and 13.87 mM, resulting in a net increase in β-cell volume when glucose levels are in the range 5.55–13.87 mM.

## 2.3 Model calibration

The model has a total of 25 parameters. All the model parameters and the method used to set their values are listed in Table 1. The parameters describing the flow rate between compartments ($Q$) and the medium volumes in the liver and pancreas compartments ($V_{m,liver}$ and $V_{m,pancreas}$, respectively) were set to the actual MPS operating conditions during the experiment. The volume of HepaRG cells in the liver compartment ($V_{HepaRG,spheroids}$) was estimated based on the number of HepaRG/HHSteC spheroids in the co-culture (40) and the number of HepaRG cells per spheroid (24,000), assuming an average hepatocyte volume of $3.4 \cdot 10^{-9}$ cm$^3$ as reported in [61]. Similarly, the volume of pancreatic β cells at the start of the co-culture ($V_{\beta,islets}(0)$) was approximated from the number of pancreatic islets (10), assuming that the proportion of β cells per islet is approximately 50%, and that each islet was spherical with a diameter of 150 μm [62].

A subset of parameters characterizing insulin secretion and changes in volume of pancreatic β cells ($EC50_I$, $d_0$, $r_1$ and $r_2$) were defined according to values reported in previous studies [47]. The parameters that define normoglycemic concentrations in the co-culture ($G_{normo}$) and $\tau_{slow}$ were approximated based on physiological considerations about the MPS system in the study, as previously described (Section 2.2). The remaining 12 parameters were estimated on an experiment-specific basis. Four of these 12 model parameters represent offsets in glucose

**Table 1. Parameters in the computational model.** Parameters specified as constant were not included in the parameter estimation routine. The estimated parameter values for each MPS experiment are listed in S3 Table.

| Parameter | Description | Unit | Estimation/reference |
|---|---|---|---|
| **Operating conditions** | | | |
| $V_{m,liver}$ | Volume of co-culture medium in the liver compartment | L | Set based on MPS operating conditions ($3 \cdot 10^{-4}$, constant) |
| $V_{m,pancreas}$ | Volume of co-culture medium in the pancreas compartment | L | Set based on MPS operating conditions ($3 \cdot 10^{-4}$, constant) |
| $V_{HepaRG, spheroids}$ | Total volume of HepaRG cells in the MPS | L | Set based on MPS operating conditions ($3.4 \cdot 10^{-6}$, constant) |
| $V_{sample,liver}$ | Volume of co-culture medium collected from the liver compartment in each sample | L | Set based on MPS operating conditions ($1.5 \cdot 10^{-5}$, constant) |
| $V_{sample, pancreas}$ | Volume of co-culture medium collected from the pancreas compartment in each sample | L | Set based on MPS operating conditions ($1.5 \cdot 10^{-5}$, constant) |
| $Q$ | Flow rate between culture compartments | L/h | Set based on MPS operating conditions ($2.96 \cdot 10^{-4}$, constant) |
| $G_{dose}$ | Glucose dose in each media exchange | mmol/L | Set based on MPS operating conditions (11, 5.5 and 2.8 for hyper-, normo- and hypoglycemia respectively, constant) |
| **HepaRG/HHSteC spheroids** | | | |
| $E_{G0}$ | Insulin-independent glucose disposal rate | 1/h | Estimated from data |
| $CL_{I,spheroids}$ | Insulin elimination rate constant | 1/h | Estimated from data |
| *Insulin resistance (slow model)* | | | |
| $G_{normo}$ | Glucose concentration for normoglycemia | mmol/L | Set based on physiological considerations (5.5, constant) |
| $S_{I0}$ | Insulin sensitivity at the start of the co-culture | L/mIU/h | Estimated from data |
| $I_{max,Si}$ | Maximal fractional reduction of insulin sensitivity | | Estimated from data |
| $EC50_{Si}$ | Value of time integral of excess glucose providing half of the maximal fractional reduction. | mmol·h/L | Estimated from data |
| **Pancreatic islets** | | | |
| $\sigma_{max}$ | Insulin secretion rate of the β cells at the start of the co-culture | mIU/L/h | Estimated from data |
| $\alpha$ | Parameter defining the sigmoidal dependence of the insulin secretion capacity on time | $h^2$ | Estimated from data |
| $EC50_I$ | Glucose concentration resulting in half-of-maximum response to insulin of the β cells | mmol/L | From literature [47] (7.86, constant) |
| *β-cell dynamics (slow model)* | | | |
| $d_0$ | Rate of β-cell death at zero glucose | 1/h | From literature [47] ($2.5 \cdot 10^{-3}$, constant) |
| $r_1$ | Rate constant that determines the dependence of the replication and apoptosis rates on glucose | L/mmol/h | From literature [47] ($6.3 \cdot 10^{-4}$, constant) |
| $r_2$ | Rate constant that determines the dependence of the replication and apoptosis rates on glucose | $L^2$/mmol$^2$//h | From literature [47] ($3.24 \cdot 10^{-5}$, constant) |
| $k_v$ | Scaling parameter for the rate of change of β-cell number | | Estimated from data |
| $\tau_{slow}$ | Constant for time averaging of glucose concentration | h | Estimated from literature [54,57–60] (500, constant) |
| **Experimental errors** | | | |
| $\Delta G_{d1}$ | Offset in glucose concentration related to co-culture media exchange in the GTT initiated at day 1 (GTT d1-3) | mmol/L | Estimated from data |
| $\Delta G_{d13}$ | Offset in glucose concentration related to co-culture media exchange in the GTT initiated at day 13 (GTT d13-15) | mmol/L | Estimated from data |
| $\Delta I_{d1}$ | Offset in insulin concentration related to co-culture media exchange in in the GTT initiated at day 1 (GTT d1-3) | mIU/L | Estimated from data |
| $\Delta I_{d13}$ | Offset in insulin concentration related to co-culture media exchange in in the GTT initiated at day 13 (GTT d13-15) | mIU/L | Estimated from data |

($\Delta G_{d1}$, $\Delta G_{d13}$) and insulin concentrations ($\Delta I_{d1}$, $\Delta I_{d13}$) related to co-culture media exchanges at the beginning of each GTT. Parameter estimation was performed using nonlinear optimization, by finding parameter values that provided an acceptable agreement with the experimental

data according to the following cost function:

$$V(p) = \sum_i \sum_t \frac{(y_i(t) - \hat{y}_i(t,p))^2}{SEM_i(t)^2}$$

where $i$ is summed over the number of experimental time-series for the given experiment $y_i(t)$ and $\hat{y}_i(t,p)$ represents the model simulations and $p$ the model parameters. SEM denotes the standard error of the mean and $t$ the measured time points in each time-series. Therefore, the value of the cost function $V(p)$ was calculated over all measured time points for all time-series considered in the optimization. We used a simulated annealing approach [63] to find the set of acceptable parameters that provided good agreement with the data according to a statistical $\chi^2$ test [64,65]. We chose a significance level of 0.05, and the number of degrees of freedom was set to the number of data points in the experimental data.

### 2.4 Software

Computations were carried out in MATLAB R2018b (The Mathworks Inc., Natick, Massachusetts, USA) using IQM tools (IntiQuan GmbH, Basel, Switzerland) and the MATLAB Global Optimization toolbox, as well as in Python (v 3.8.8). The freely available software WebPlotDigitizer 4.3 (https://automeris.io/WebPlotDigitizer) was used to extract the experimental data from the study by Dalla Man et al. [66]. Figures were prepared using BioRender (https://biorender.io/) and Illustrator CC 2019 (Adobe).

### 2.5 Data correction

The number of replicate platforms considered in the study was on average 5, varying between 4 and 10 across the different experiments. Due to this small sample size, we assume that the measured SEM is an underestimation of the uncertainty in the data and SEM values below 5% of the corresponding mean are considered unrealistic. To correct for such possible underestimations in data uncertainty, we set the SEM of data points with a measured SEM below 5% of their mean to the largest measured SEM value across all points in the dataset. In experiments where all datapoints had a SEM value below 5% of their mean, the SEM was changed to 10% of their mean instead. The resulting SEM values in each experiment are given as errors bars in the figures included in the main article and S2 Fig.

## 3 Results

### 3.1 The integrated experimental-computational approach

A flow chart of the steps involved in the experimental-computational approach is presented in Fig 3. First, we developed a computational model for the interplay between glucose and insulin in the HepaRG liver-islet MPS, describing fast (hours) glucose homeostasis and slow (2 weeks) changes in insulin sensitivity and β-cell dynamics (Fig 2). We limited the complexity of the model to represent only mechanisms needed to describe the experimental data in the study, thereby keeping the model's size small (12 free model parameters). A complete description of the model equations, as well as the code used for simulations, are provided in the S1 Appendix.

The next step was to calibrate the model on an experiment-specific basis. To perform this calibration, the parameters were estimated using the available data from the corresponding experiment. These data varied among the seven experiments (S1 Table), and comprised combinations of the following time-series measurements: 1) glucose and insulin concentrations during GTTs in co-cultures under two different glycemic regimes (hyper-, and/or

normoglycemia) and 2) glucose concentrations during GTTs in HepaRG single-liver cultures under hyperglycemia. The model development and calibration steps were executed in an iterative manner, allowing us to modify the model with each iteration until it was able to accurately describe the calibration data. The final model resulting from this iterative process is the one proposed in this paper. Last, we evaluated this model by testing its ability to predict data not considered during calibration in two of the seven experiments.

## 3.2 Quantitative analysis of the mechanisms behind impaired glucose homeostasis over 15 days of co-culture

We applied the computational model to quantitatively describe the physiological processes behind impaired glucose homeostasis in the liver-islet co-cultures. We calibrated the model using data from an experiment where both liver-islet and single-liver cultures were exposed to hyperglycemic conditions mimicking high plasma glucose in T2DM (experiment 2 [18]). Fig 4A–4C shows a comparison between the model simulations and the experimental measurements used for calibration, which included time-series data of both glucose and insulin concentrations during GTTs in the co-cultures (Fig 4A and 4C respectively, red markers), and glucose concentration in the single-liver cultures (Fig 4B, blue markers). The estimated parameter values provided an acceptable agreement between the model simulations (Fig 4A–4C, lines) and the data (Fig 4A–4C, markers), as determined by $\chi^2$ statistics (S2 Table). This agreement is statistically supported by the fact that the model passes a $\chi^2$ test at significance level $\alpha = 0.05$, with a value of the cost for the optimal parameter set $p_{opt}$ lower than the $\chi^2$-threshold ($V(p_{opt}) = 21.62 < 31.41$).

As shown in Fig 4A, glucose levels in the liver-islet co-culture during the GTT initiated at day 1 (GTT d1-3) reached a glucose concentration of 7.45 ± 0.63 mM, within the normoglycemic range (3.9–7.8 mM) [18], within eight hours. On the contrary, we observed a slower glucose consumption after 13 days of co-culture, with a glucose concentration of 8.77 ± 0.74 mM eight hours after the start of the GTT. In contrast, in single-liver cultures, glucose levels remained within hyperglycemia for the entire co-culture period (Fig 4B). These changes in glucose dynamics were accompanied by a decrease in insulin concentration levels over time, as seen in Fig 4C. We used the modelling approach to infer variables that could mechanistically describe the physiological changes underlying these alterations in glucose regulation and β-cell function. According to the model, insulin sensitivity in the HepaRG/HHSteC spheroids decreases progressively as they are exposed to hyperglycemic periods during the co-culture (Fig 4E). The sustained hyperglycemic levels are quantified in the model as the integral of excess glucose (difference between glucose levels in the co-culture media and 5.5 mM) over time (Fig 4D). This decline in insulin sensitivity potentially leads to reduced glucose utilization by the HepaRG/HHSteC spheroids, resulting in higher daily glucose levels over time (Fig 4F). In turn, the model suggests an increase in the number of β cells to compensate for the rise in glucose levels, and therefore β-cell volume increases (Fig 4G). However, besides this adaptation in β-cell volume, the secretion capacity of the individual β cells decays (Fig 4H). These combined effects result in a decline in circulating insulin levels over the 15-day co-culture period, in agreement with the experimental measurements (Fig 4C). The model predicted a decrease in insulin sensitivity for all acceptable parameters, with an average and maximal decrease in insulin sensitivity of 7.5% and 66% of its initial value, respectively, across all parameter values.

We estimated parameters for each MPS experiment individually to fit the corresponding measurements of glucose and insulin, and achieved an acceptable agreement between the model simulations and the experimental data (S2 Table). A comparison between the model simulations and the experimental measurements for the seven experiments included in the study is shown in S2 Fig. The estimated parameter values for each MPS experiment are listed in S3 Table.

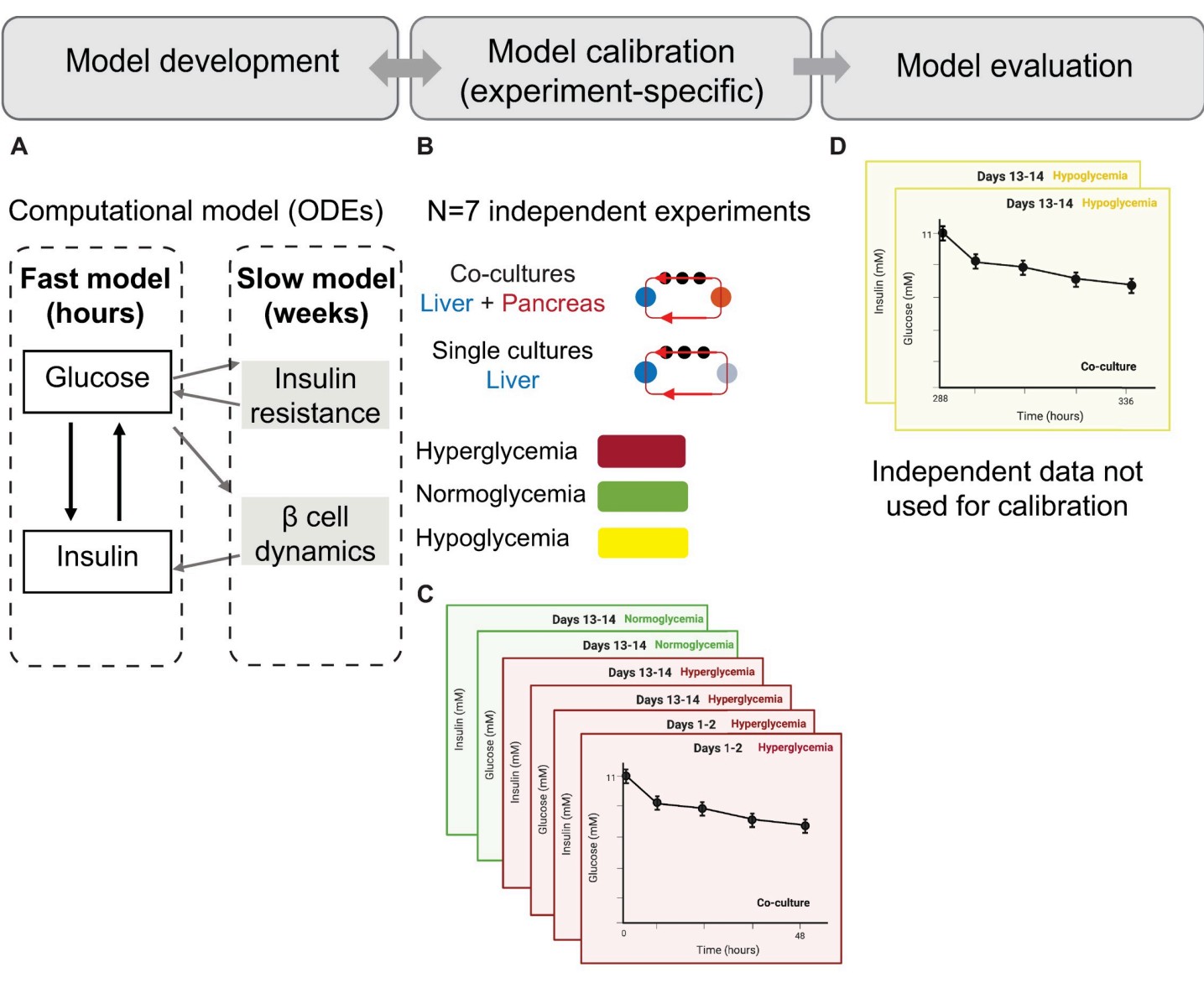

**Fig 3. Flow chart illustrating the steps in the modelling approach. (A)** Model development: The computational model includes a fast component governing daily interactions between glucose and insulin, and a slow component describing the development of insulin resistance and pancreatic β-cell compensation. **(B)** Model calibration: The parameters in the model were estimated individually for each of the seven experiments (N = 7). Experimental data used for calibration **(C)** includes time-series measurements of glucose and insulin from both co-cultures and single-liver cultures, as well as from co-cultures exposed to different glycemic regimes (hyper- and normoglycemia). These measurements were acquired during 48-hour GTTs initiated at day 1 or day 13. **(D)** Model evaluation: The model was evaluated against independent data not used in the calibration step. This evaluation was performed in two of the seven experiments (N = 2) in which we acquired additional *in vitro* measurements to compare against the model predictions.

### 3.3 Investigating the effect of glycemic regimes on glucose metabolism

To further investigate the effect of glycemic levels on HepaRG/HHSteC spheroid-pancreatic islet cross-talk, we applied the computational model to leverage data from experiments under varying glycemic conditions. We performed *in vitro* experiments where co-cultures were exposed to both normo- and hyperglycemic glucose levels emulating healthy and T2DM conditions, respectively.

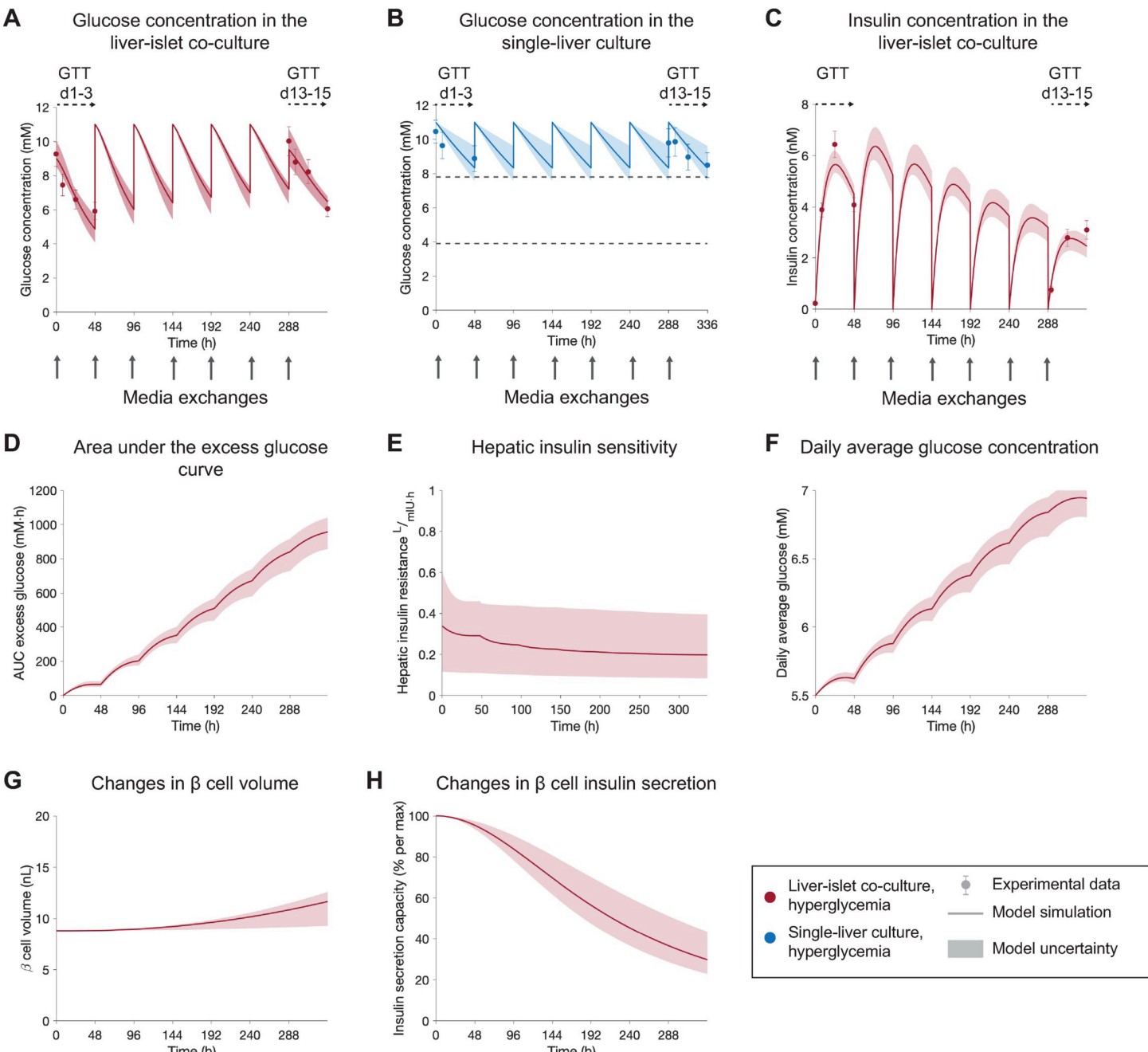

**Fig 4. Model-based analysis of the physiological processes behind impaired glucose homeostasis in the co-culture.** The results correspond to experiment 2 [18]. **A-C**: Comparison between experimental measurements (markers) and model simulations (lines) of glucose concentration in the co-cultures (**A**), in single-liver cultures (**B**), and insulin concentration in the liver-islet co-cultures (**C**). Co-cultures were exposed to hyperglycemic medium (11 mM glucose) at each media exchange (grey arrows). Experimental time-series were acquired during GTTs initiated at day 1 (GTT d1-3) and day 13 (GTT d13-15). The mechanistic variables inferred by the model, which can explain the experimental data in (A-C) are shown in (D-H). (D) Time-integral or area under the curve (AUC) of excess glucose (i.e. difference between glucose levels in the co-culture media and 5.5 mM) over time. This accounts for the effect of exposing the HepaRG/HHSteC spheroids to periods of hyperglycemia during the co-culture time, with the associated decrease in hepatic insulin sensitivity (**E**) and rise in daily average glucose levels (**F**). Changes in β-cell insulin-producing capacity predicted by the model are caused by an increase in pancreatic β-cell volume (**G**) and a decay in the individual secretion capacity of β cells over time (**H**). Model uncertainty is depicted as shaded areas in panels **A-H**. Data in panels **A-C** are presented as mean ± SEM, n = 5. n: Number of platform replicates included in the experiment.

By using the computational model, we sought to interpret the experimental results in relation to the changes in pancreatic β-cell function and impaired glucose tolerance.

We assessed how repeated exposure of the co-cultures to two different glycemic conditions (hyper- and normo-glycemia) impacted their response to a glucose load. To do so, the co-cultures were exposed to either hyper- or normoglycemic glucose levels (11 mM or 5.5 mM glucose, respectively) at each medium exchange during the first 13 culture days. In the hyperglycemic co-cultures, we performed a GTT at day 1 (GTT d1-3, Fig 5A and 5B) to evaluate glucose tolerance at the beginning of the co-culture. We then performed GTTs on day 13 (GTT d13-15, Fig 5A and 5B) on both hyper- and normoglycemic co-cultures, to establish a comparison with the response from the initial GTT for both glycemic regimes. The GTT initiated at day 1 was only performed in the co-cultures maintained under hyperglycemia, to avoid exposure of HepaRG/HHSteC spheroids in normoglycemia to high glucose levels in the beginning of the co-culture.

To calibrate the model, we estimated the model parameters using data from both normo- and hyperglycemic conditions simultaneously. Only the parameters corresponding to glucose offsets due to media exchanges ($\Delta G_{d1}$, $\Delta G_{d13}$) were estimated for each condition independently. Furthermore, the parameter describing glucose dosing to the system ($G_d$) was set accordingly for each condition (11 mM or 5.5 mM). Analysis of the model simulations indicate that HepaRG/HHSteC spheroids exposed to normoglycemic conditions over the co-culture period exhibited higher insulin sensitivities than those maintained under hyperglycemia (Fig 5C). Thus, although insulin levels in hyperglycemic conditions during the GTT performed at day 13 were higher than those under normoglycemia (by 4-fold 24 h after the start of the GTT, Fig 5B), glucose levels were comparable for both glycemic regimes (Fig 5A). These results are in line with our previous observations on the possible development of insulin resistance due to exposure to high glucose levels (Fig 4). The observed differences in insulin secretion between hyper- and normoglycemia (Fig 5B) could also be reproduced by the model. These differences could be explained by the bell-shaped relationship between average glucose levels in the co-culture and the net change of β-cell volume (Eq 10). In the experiments under hyperglycemia, daily average glucose levels varied between 5.5 mM and 7.2 mM over the the 15-day co-culture period (Fig 5D). Thus, the model predicted a net increase in β-cell volume (Fig 5E), as these values lie within the range of glucose levels for increased rate of change of β-cell number (i.e. replication minus apoptosis) suggested by Topp et al. [52], which is set to 5.55–13.87 mM based on the study from Topp et al. [47]. On the contrary, daily average glucose levels in co-cultures maintained under normoglycemia were within the range of 5.0–5.5 mM (Fig 5D) postulated to lead to a decrease in β-cell volume (Fig 5E) and the resulting decay in insulin secretion compared to hyperglycemia (Fig 5B).

## 3.4 Assessing the predictive capabilities of the computational model

**3.4.1 Prediction of glucose and insulin responses under hypoglycemia.** To evaluate our computational model, we assessed whether it was able to predict data not employed during calibration (last step in Fig 3). To do so, we applied an experiment-specific model calibrated to both normo- and hyperglycemic conditions simultaneously (experiment 3, Fig 5) to simulate hypoglycemia *in silico*. We then performed the corresponding *in vitro* MPS experiment, where the co-cultures were exposed to hypoglycemic glucose levels (2.8 mM) at each medium exchange during the first 13 culture days, followed by a GTT at day 13. To account for experimental uncertainties in the glucose dose administered to the system at the start of the GTT, we allowed the value of the glucose dose to vary within the measured range of SEM (± 0.85 mM) when computing the model predictions.

The predictions of glucose and insulin responses during the GTT initiated at day 13 (GTT d13-15, Fig 6B and 6D, shaded areas) were in good agreement with the experimental data (Fig

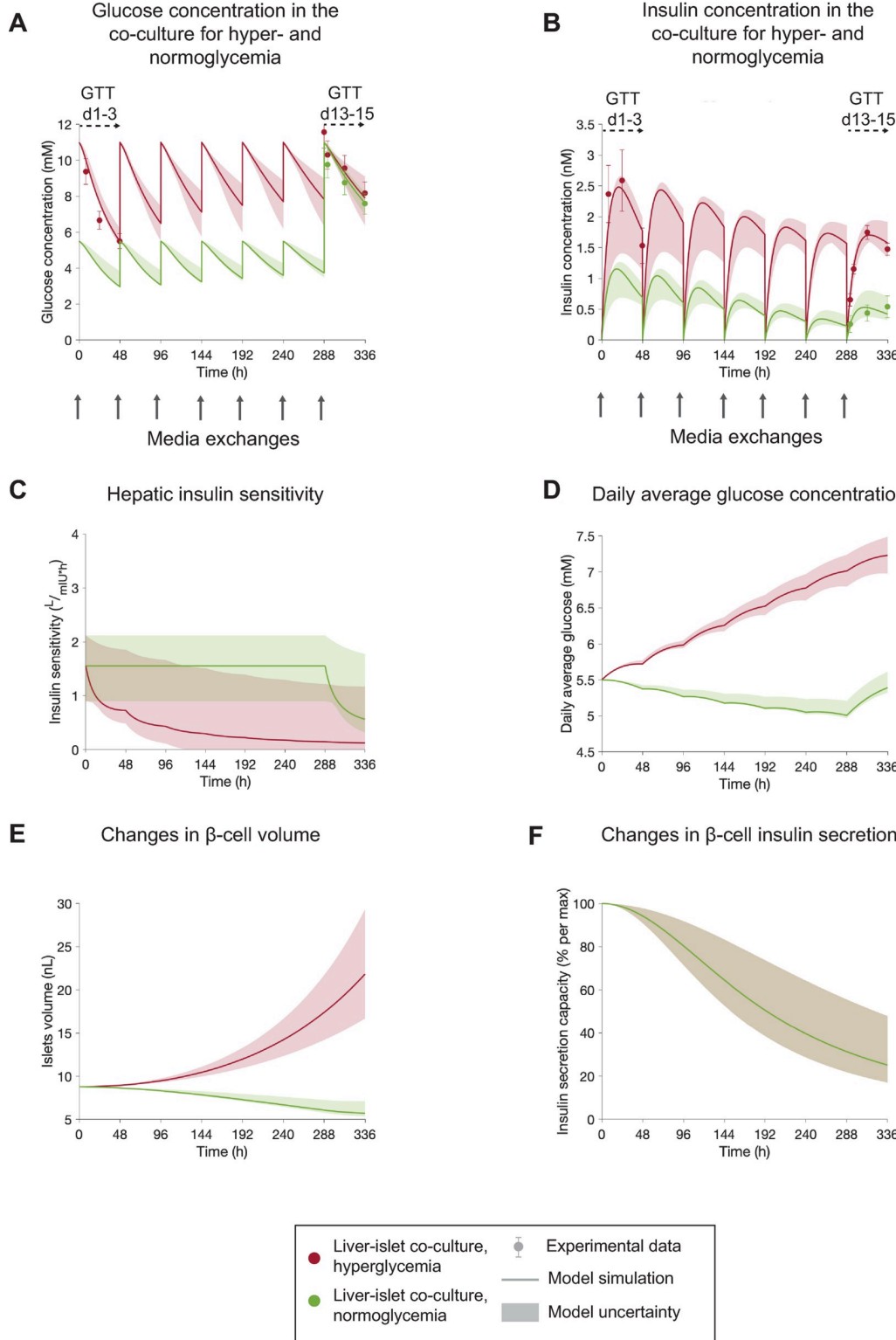

**Fig 5. The computational model can explain impaired glucose homeostasis and β-cell dysfunction under varying glycemic conditions in HepaRG liver-islet MPS.** The results correspond to a single experiment (experiment 3). **A-B:** Comparison between experimental measurements (markers) and model simulations (lines) of glucose concentration (**A**), and insulin concentration (**B**) over the 15 days of liver-islet co-culture. Co-cultures were exposed to either a hyperglycemic (11 mM glucose, red) or normoglycemic medium (5.5 mM glucose, green) in each media exchange (grey

arrows) between days 1 and 13. Experimental time-series were acquired during GTTs initiated at day 1 (GTT d1-3) in hyperglycemic co-cultures and day 13 (GTT d13-15) in both hyper- and normoglycemic co-cultures. The model predicts decreased insulin sensitivity (i.e. increased insulin resistance) in HepaRG/HHSteC spheroids from hyperglycemic co-cultures, compared to those under normoglycemia **(C)**. The predicted differences in daily glucose levels over the co-culture time between both glycemic regimes **(D)** lead to different trends in β-cell volume changes **(E)** due to the bell-shaped relationship between average daily glucose and net β-cell growth rate in the model. The changes in β-cell insulin secretion predicted by the model do not depend on the glycemic levels and therefore are the same for both glycemic regimes (i.e. overlap in (**F**)). Model uncertainty is depicted as shaded areas in panels **A-E**. Data in panels **A-B** are presented as mean ± SEM, n = 5. n: Number of platform replicates included in the experiment.

6B and 6D, markers) according to a statistical $\chi^2$ test at significance level $\alpha = 0.05$ ($V(p_{opt})$ = 12.79 < 14.07). The validation was performed against glucose and insulin experimental measurements, but the additional mechanistic variables provided by the model, such as β-cell volume, insulin secretion capacity and insulin resistance were not validated against experimental measurements.

**3.4.2 Prediction of long-term changes in glucose and insulin responses.**    Next, we assessed the ability of the computational model to predict long-term changes in glucose and insulin responses in the liver-islet co-culture over time, and how these were influenced by the operating conditions in the MPS (i.e. the flow rate distribution on-chip and the medium volume in each culture compartment). To that end, in one experiment (experiment 3), we measured glucose and insulin concentrations periodically 48 hours after each media exchange between days 3 and 13 of the co-culture. In addition, to characterize the effect of the MPS operating conditions on the observed dynamics, we collected glucose and insulin samples from each culture compartment (liver and pancreas). The model was calibrated using glucose and insulin data acquired during GTTs initiated at day 1 and 13 (GTT d1-3 and GTT d13-15, Figs 4 and 7C and 7F). In the experimental data used for calibration, the concentrations were measured by pooling samples of 15 μL from each compartment (i.e. measuring the average concentration of the two compartments), as opposed to the compartment-wise measurements acquired for the evaluation dataset. The model predictions (Fig 7A, 7B, 7D and 7E, shaded areas) showed good visual agreement with the experimental measurements in terms of both temporal evolution of glucose and insulin dynamics along the co-culture, and the glucose concentration levels in each culture compartment (Fig 7A, 7B, 7D and 7E, markers). The computational model also captured the greater insulin concentration in the pancreas compartment compared to that in the liver compartment. However, the model predicted insulin concentration between days 5 and 9 that were slightly larger than those found experimentally. Nevertheless, given that the model is not calibrated using these data, there is still a good agreement between the model predictions and the experimental data, and the prediction error is comparable to the size of the variability in the experimental measurements.

## 3.5 Translation to *in vivo* responses in humans

Following evaluation, we investigated whether a model-based scaling strategy could translate the MPS results from *in vitro* to *in vivo*. We established an upscaling approach that involves extrapolation of the following model parameters: volumes of the organoids, flow rate and volume of co-culture medium in the compartments. To upscale the volume of the organoids to human proportions, we multiplied the total volume of both HepaRG cells ($V_{HepaRG,spheroids}$) and pancreatic β cells ($V_{\beta,islets}(0)$) by the 100,000 factor applied in the miniaturization to MPS. The total volume of co-culture medium was scaled to 3 L, under the assumption that the blood volume in humans is approximately 5.1 L [43] with a plasma proportion of 58% [67]. This volume was distributed equally between both tissue compartments, resulting in a medium volume

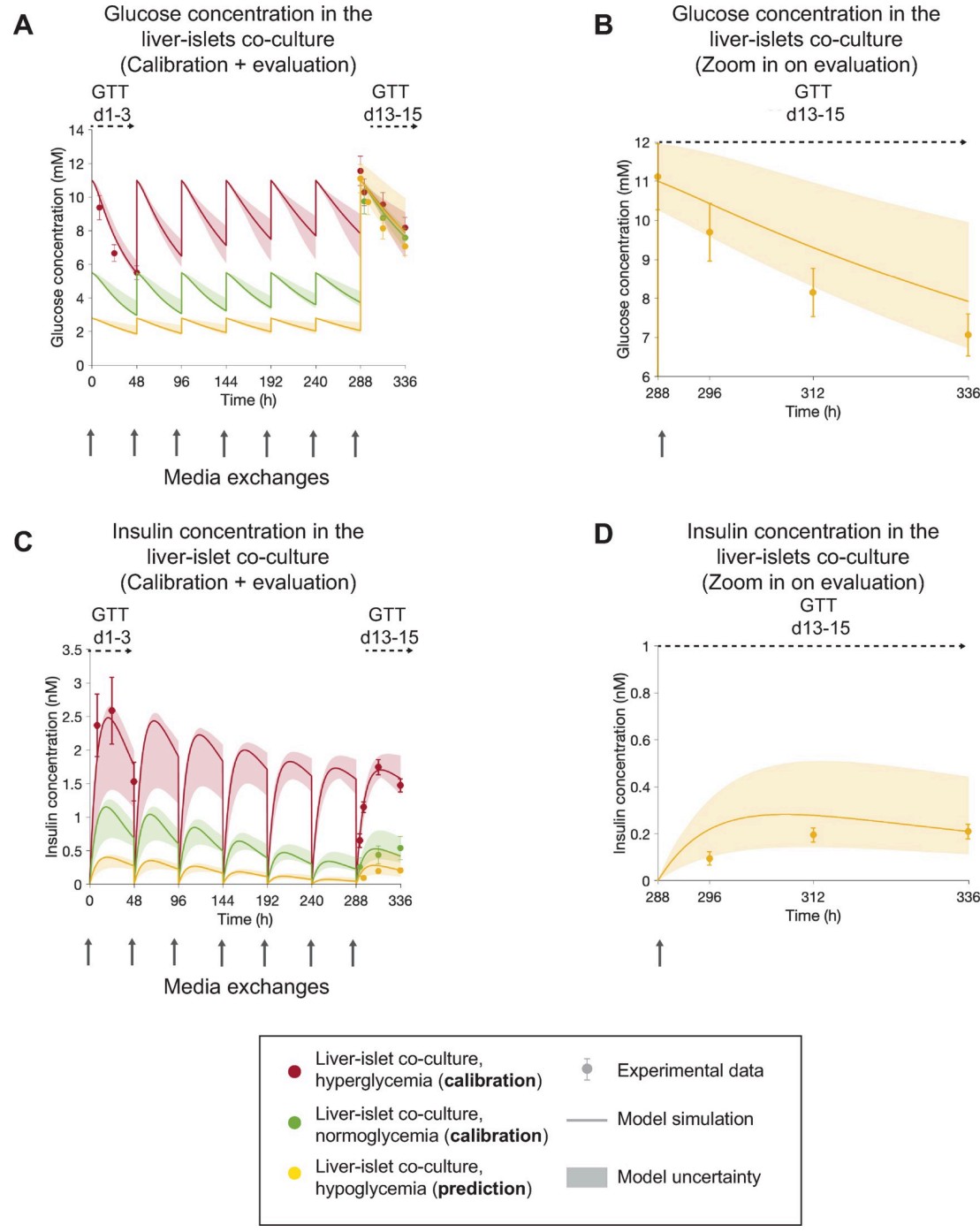

**Fig 6. The model can predict GTT responses in the end of the liver-islet co-culture following repeated exposure to hypoglycemic media concentrations.** The results correspond to a single experiment (experiment 3). In **A,C,** experimental measurements of glucose and insulin concentrations (markers) are compared to model simulations for hyper- and normo-glycemia (red and green, lines) and model predictions for hypoglycemia (yellow lines) over the 15 days of co-culture. Co-cultures were exposed to either a hyperglycemic (11 mM glucose, red), normoglycemic (5.5 mM glucose, green) and hypoglycemic (2.8 mM glucose, yellow) media in each media exchange (grey arrows) between days 1 and 13. Experimental time-series were acquired during GTTs initiated at day 1 (GTT d1-3) in hyperglycemic co-cultures and day 13 (GTT d13-15) in all co-cultures (hyper-, normo- and hypoglycemic). For a clearer comparison between the model predictions and the corresponding experimental data for the evaluation part, panels **B** and **D** zoom in on the GTT performed at day 13 for the hypoglycemic arm—glucose (**B**) and insulin (**D**). Model uncertainty is depicted as shaded areas in **A-D**. Data in panels **A,C** are presented as mean ± SEM, n = 5. n: Number of platform replicates included in the experiment.

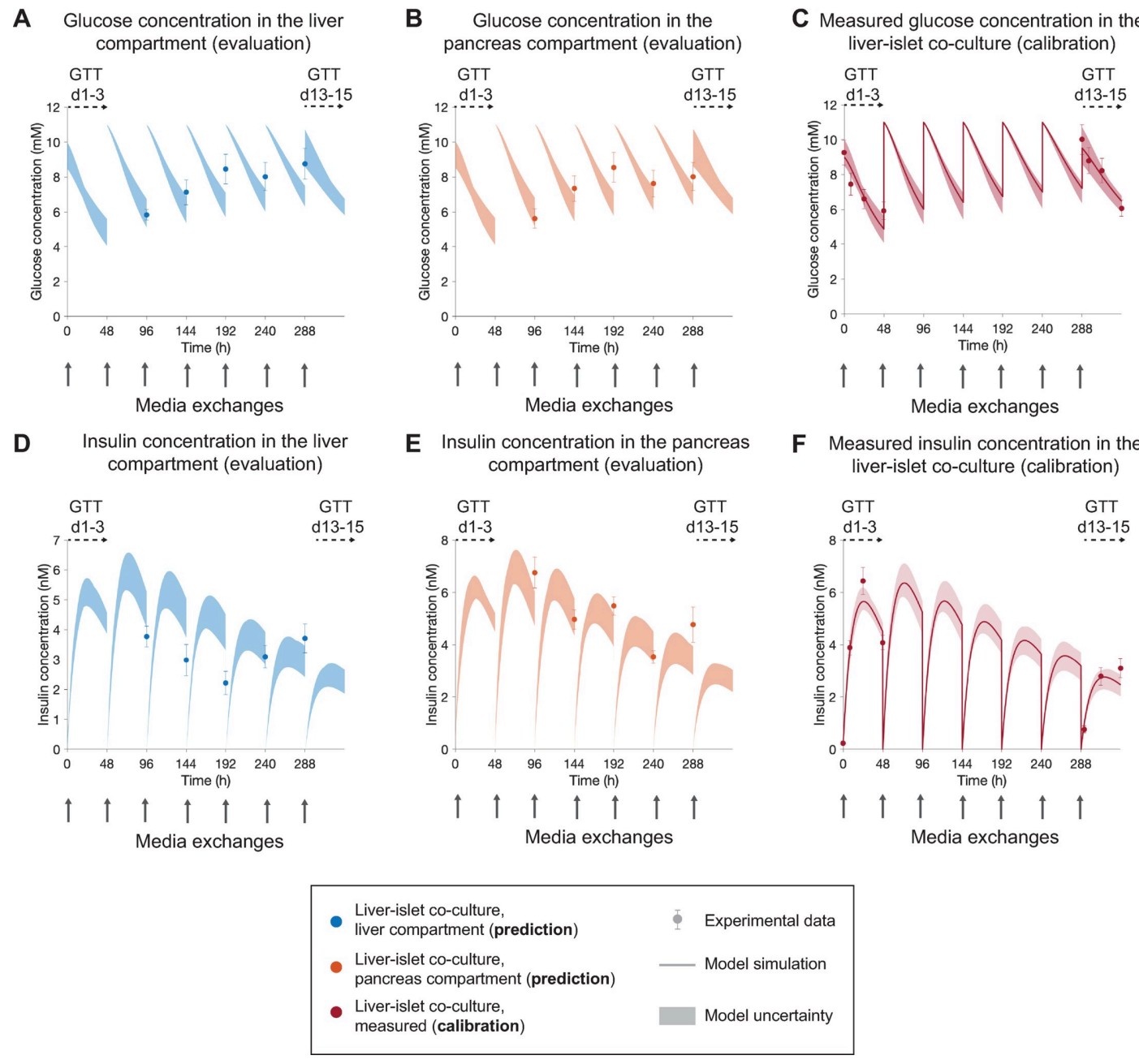

**Fig 7. Model predictions of glucose and insulin concentrations in each organ compartment.** The results correspond to a single experiment (experiment 2). **A,B** Comparison between model predictions of glucose in the liver and islets compartments, respectively, and the corresponding experimental data (markers). **D,E** Comparison between model predictions of insulin in the liver and islets compartments, respectively, and the corresponding experimental data (markers). Co-cultures were exposed to hyperglycemic medium (11 mM glucose) in each media exchange (grey arrows). The calibration data **(C,F)** consisted of glucose and insulin measurements acquired during GTTs initiated at day 1 (GTT d1-3) and day 13 (GTT d13-15). In the calibration data **(C,F)**, glucose and insulin concentrations were measured by pooling samples of 15 μL from each compartment (i.e. measuring the average concentration of the two compartments), while concentrations in the evaluation dataset **(A,B,D,E)** were measured independently in each culture compartment (liver or pancreas). Data points for evaluation were acquired 48 hours following each media exchange between days 3 and 13. Model uncertainty is depicted as shaded areas in **(A-F)**. Data are presented as mean ± SEM, n = 10. n: Number of platform replicates included in the experiment.

of 1.5 L in each compartment. The flow rate ($Q$) was then set to achieve a media turnover time of 5 min, as observed in humans [68].

As a proof-of-concept demonstration, we tested the scaling strategy in one of the experiments (experiment 1). We first calibrated the original model using data of glucose and insulin during GTTs initiated at day 1 (GTT d1-3) in both liver-islet co-cultures and single-liver cultures (S2A–S2C Fig, markers). The parameters describing the operating conditions in the system were then modified according to the proposed upscaling approach. Additionally, to account for glucose consuming organs other than the liver (i.e. muscle, adipose tissue, brain and kidneys), we increased the glucose uptake rate of the liver. More specifically, we multiplied both the insulin-independent glucose disposal rate ($E_{G0}$) and insulin sensitivity ($S_I$) by a factor of 2.22, assuming that the liver is responsible for approximately 45% of the total postprandial glucose uptake in humans [69,70]. Similarly, the insulin elimination rate constant of the liver ($CL_{I,HepaRG}$) was doubled under the assumption that the liver stands for approximately 50% of the total insulin clearance [71]. Endogenous glucose production was kept to zero for the translation, since this term was considered to be negligible for the HepaRG/HHSteC spheroids. The changes in model parameters for translation from the HepaRG liver-islet MPS to human are listed in Table 2.

After these parameter changes, the temporal dynamics of the glucose and insulin responses predicted in the model were significantly faster than in the MPS and in agreement with those found in humans [66] (S3C and S3F Fig). However, the glucose uptake predicted by the model (S3C Fig, shaded area) was larger than the *in vivo* measurement (S3C Fig, markers). Furthermore, the predicted insulin concentrations (S3F Fig, shaded area) were one order of magnitude higher than the ones measured *in vivo* (S3F Fig, markers). The peak insulin concentration in the model predictions ranged between 1.8 and 2.0 nM, while the corresponding *in vivo* value was 0.34 nM. Because insulin concentrations measured in the MPS were also one order of magnitude higher than those reported in human studies [66,72,73], we hypothesize that the increased insulin concentrations might be due to impaired insulin clearance by the HepaRG/HHSteC spheroids and/or enhanced insulin secretion by the pancreatic islets compared to the *in situ* case. To investigate the first hypothesis, we translated the model-based $CL_{I,HepaRG}$ to human hepatic insulin elimination rate constant, using Eq. S5 (S1 Appendix) and the corresponding parameter values in Table 2. The resulting hepatic insulin elimination rate constant, $CL_{human}$ = 4.04 (1/h), is 4.23 times lower than the reported value in humans [66]. Therefore, we increased the value of $CL_{I,HepaRG}$ accordingly to account for this potential effect. Enhanced islet insulin secretion, on the other hand, can possibly be due to the long-term exposure of the pancreatic islets to a hyperglycemic medium over the co-culture time, leading to overstimulation of insulin release. While our experimental results indicate enhanced GSIS values for islets that have been cultured under hyperglycemia, in comparison to those that have been cultured

**Table 2. Extrapolation of parameter values in the computational model to perform *in vitro* to *in vivo* translation.** The results correspond to experiment 1.

| Parameter | Description (units) | Value | |
|---|---|---|---|
| | | ***In vitro* (MPS)** | **Translation to *in vivo* (human)** |
| $V_{HepaRG,spheroids}$ | Total volume of HepaRG cells (L) | $3.4 \cdot 10^{-6}$ | 0.34 |
| $V_{\beta,islets}(0)$ | Total volume of pancreatic β cells at the beginning of the co-culture (L) | $8.8 \cdot 10^{-9}$ | $8.8 \cdot 10^{-4}$ |
| $V_{m,liver}$ | Volume of co-culture medium in the liver compartment (L) | $3 \cdot 10^{-4}$ | 1.5 |
| $V_{m,islets}$ | Volume of co-culture medium in the islets compartment (L) | $3 \cdot 10^{-4}$ | 1.5 |
| Q | Flow rate between culture compartments (L/h) | $2.96 \cdot 10^{-4}$ | 35.5 |
| $E_{G0}$ | Hepatic insulin-independent glucose disposal rate (mmol/L/h) | 1.47 | 3.25 |
| $S_{I0}$ | Hepatic insulin sensitivity at the start of the co-culture (L/mIU/h) | $5 \cdot 10^{-3}$ | $1.1 \cdot 10^{-2}$ |
| $\sigma_{max}$ | Maximal insulin secretion rate per unit volume of β cells (mIU/L/h) | $6 \cdot 10^{6}$ | $6 \cdot 10^{6}$ |
| $CL_{i,hep}$ | Hepatic insulin elimination rate constant (1/h) | 17.81 | 150.67 |

under normo-and hypo-glycemia (S1 Fig), we currently lack a quantitative measure of the comparison between these GSIS values and their human counterparts. We therefore halved the maximal insulin secretion rate of the β cells ($\sigma_{max}$) to achieve good visual agreement with the insulin concentration measured *in vivo*. The resulting glucose and insulin responses predicted by the computational model (Fig 8C and 8F, shaded areas) agree well with the measured ones in humans (Fig 8C and 8F, markers), even though the predicted glucose concentration decreases to values below normoglycemia as endogenous glucose production from the HepaRG/HHSteC spheroids cells is neglected in our model.

## 4 Discussion

This study demonstrated the potential of applying computational modelling in combination with MPS to augment *in vitro* investigations of glucose metabolism and allow translation to humans. We constructed a computational model of glucose homeostasis in a previously developed HepaRG liver-pancreas MPS [18]. After calibration, the model was able to replicate glucose and insulin responses under both healthy glucose levels and high plasma glucose mimicking T2DM (Fig 5). To demonstrate the predictive power of the model, we evaluated it on measurements not considered for calibration. The model could correctly predict the response of the MPS to a hypoglycemic regime (Fig 6), and the long-term dynamics of glucose and insulin over 15 days of co-culture (Fig 7). Last, we have shown that the model is able to translate *in vitro* glucose and insulin responses in the MPS to humans, showing good agreement with reported data on meal responses from healthy subjects [66] (Fig 8).

The mechanistic, computational model presented in this study aims at describing the physiology in the MPS, encapsulating mechanisms underlying glucose regulation and disease progression in T2DM (insulin resistance and $\beta$-cell adaptation). To date, studies combining computational models and MPS have mainly applied PK [28–31] and PKPD [32,33,74] approaches, and have not included experimental measurements characterizing the cross-talk between liver and pancreas. Since glucose homeostasis relies on a feedback loop involving storage and release of glucose by the liver in response to glucose-regulated insulin secretion from the pancreas, it is crucial to examine data from interconnected co-cultures that reflect this organ interplay. Moreover, the computational model in Lee et al. [41] was designed to simulate a 3-hour response to a meal, and, in contrast to our model, did not include a description of disease progression.

Several computational approaches have also been developed to describe glucose homeostasis and different aspects of T2DM in animal models. These could potentially be used in combination with preclinical animal studies, to help in data interpretation and extrapolating the results to humans. Alskär et al. [73] demonstrated that an allometrically scaled model of human glucose homeostasis [75] could reproduce glucose and insulin responses in several preclinical animal species. However, this model only describes short-term regulation of glucose homeostasis during GTT and cannot simulate the long-term pathophysiology of T2DM. In contrast, other animal-based computational models have focused on long-term changes in weight [76,77], but they do not establish a link with pathophysiological defects implicated in T2DM such as insulin resistance.

The computational model presented in our study could simultaneously describe glucose and insulin responses in the MPS under both normo- and hyperglycemic conditions representative of T2DM (Fig 5A and 5B). The fact that the model can reproduce a range of behaviors consistent with experimental observations only through estimation of the model parameters suggests the generality of the model structure, that is, the robustness of the mathematical equations. However, to further build confidence in the computational model, it is crucial to

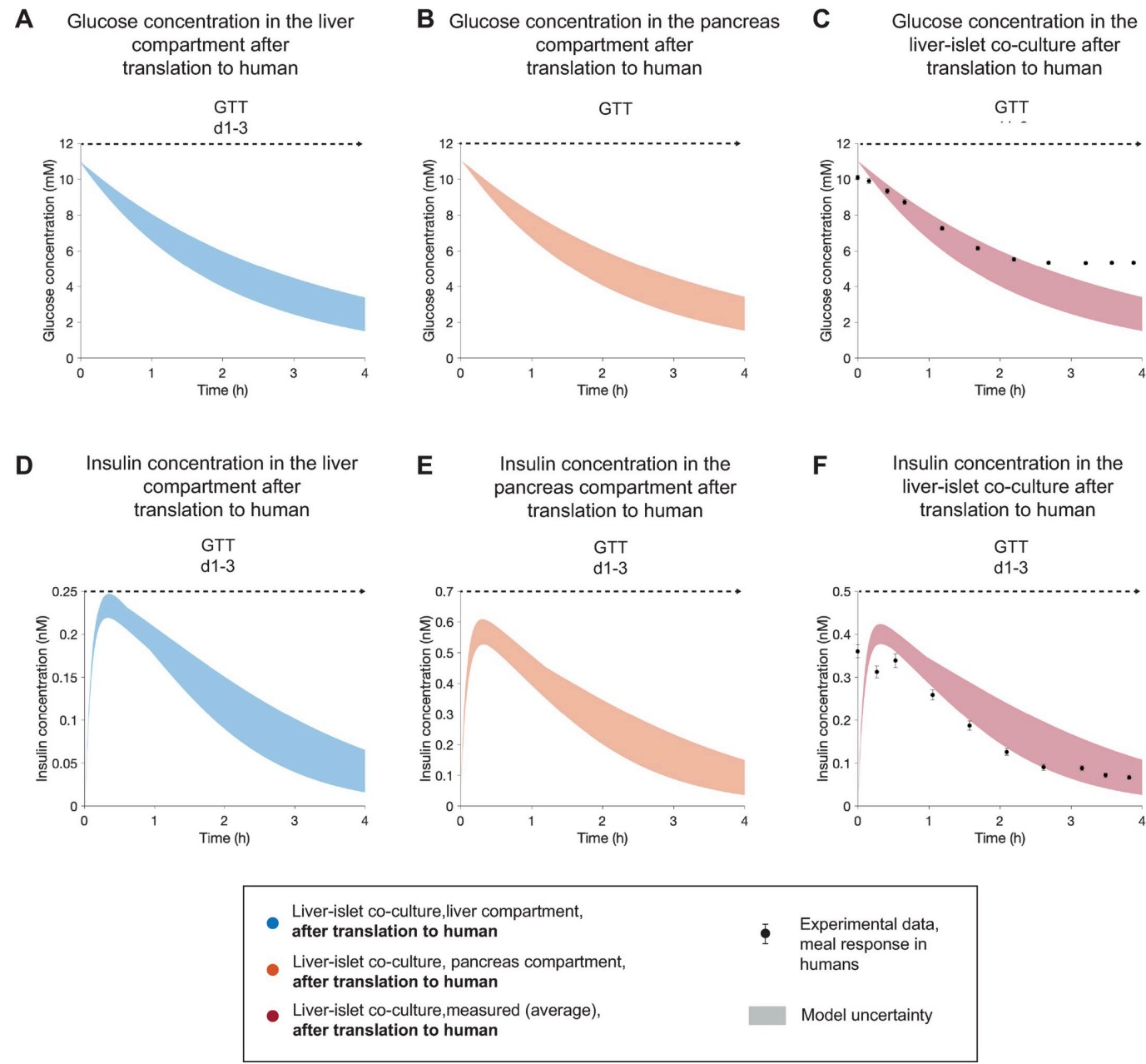

**Fig 8. Model-based extrapolation of glucose and insulin concentrations in the HepaRG liver-islet MPS to human meal responses.** The results correspond to a single experiment (experiment 1). **A,B,D,E** Model predictions of glucose (**A,B**) and insulin (**D,E**) in the liver and islets compartments after scaling to human. **C** shows the comparison between the model prediction of plasma glucose concentration in a corresponding human and experimental data of glucose response to a meal in healthy subjects [66]. The model-based prediction of the insulin response and the experimental measurements of insulin are compared in **F**. The predictions are computed for the GTT initiated at day 1 (GTT d1-3). The experimental data, reported in a previous study by Dalla Man et al. [66], were acquired in a group of 204 normal subjects. We consider the time point of peak glucose concentration in the experimental data as time = 0 h for this study, since the HepaRG liver-islet MPS lacks an intestinal compartment and glucose is administered directly to the liver and pancreas compartments. Data are presented as mean ± SEM, n = 204. Model uncertainty is depicted as shaded areas in (**A-F**).

evaluate its predictive capability against experimental data not considered during calibration [78,79]. Considering this, we first calibrated the model using MPS measurements from normo- and hyper-glycemic conditions, and subsequently evaluated the model predictions under a different glycemic regime (hypoglycemia) using data from an independent MPS

experiment. We showed that the model was able to predict glucose and insulin responses under this new glycemic regime, although it was not calibrated for this purpose (Fig 6B and 6D). In addition, we demonstrated the ability of the model to predict glucose and insulin concentrations at times not included in our experimental sampling protocol, and in both culture compartments (Fig 7A, 7B, 7D and 7E).

Our approach also offers mechanistic insight through simulation of key physiological variables that are not measured in the experiments, such as insulin resistance and the total volume of β cells in the co-culture (Figs 4 and 5). Our model predicted an increase in total β-cell volume after exposure to hyperglycemic glucose levels over 15 days of co-culture, while this volume was predicted to decrease when the pancreatic islets were exposed to normoglycemic levels (Fig 5E). These predictions follow from the previously established hypothesis that small deviations from normoglycemia cause β-cell volume increase as a feedback mechanism to reestablish glucose homeostasis [47], which stems from previous studies reporting a nonlinear variation of both β-cell replication and apoptosis rates *in vitro* [54,80]. GSIS analysis from our co-culture experiments revealed increased insulin secretion in pancreatic islets co-cultured under hyperglycemic conditions (11 mM) as compared to those in either normo- or hypoglycemic co-cultures (5.5 and 2.8 mM, respectively) (S1 Fig). This might indicate that the total volume of β cells in the end of the co-culture is larger when they have been exposed to hyperglycemic concentrations. However, GSIS measurements do not only reflect the volume of β-cells but also their individual secretion capacity. To increase confidence in this model prediction, future studies should be carried out to establish a comparison against experimentally measured β-cell volumes in the liver-islet co-culture under different glycemic levels, for instance focusing on β-cell proliferation. There may be also additional biological mechanisms governing β-cell dynamics that are not considered in the computational model. For instance, in our study the insulin secretion capacity of the β cells was assumed to decrease over time. However, previous studies have hypothesized an increase in β-cell secretion capacity via reduction in the number of ATP-sensitive $K^+$ (KATP) channels of the β cell as a compensatory mechanism to sustained, elevated glucose levels [49]. Similarly, β-cell secretion capacity might initially rise and then fall, leading to the increase and subsequent decrease in insulin levels typically observed in human and animal studies [3,49]. Experimental measurements characterizing β-cell volumes could help in elucidating the compensatory changes in both β-cell volume and function in response to hyperglycemia.

The complexity of the computational model was chosen based on the intended level of detail and the available experimental measurements. With this in mind, we only modelled the physiological mechanisms needed to simulate the data in the study. Here, the only measurements available to characterize the contribution of the HepaRG/HHSteC spheroids to glucose metabolism were glucose concentrations in the co-culture medium over time. Therefore, we established a relatively simple model of hepatic glucose metabolism that only captured net glucose fluxes between the HepaRG/HHSteC spheroids and the co-culture medium, but did not describe any intracellular fluxes. Additional experiments using isotope labeling tracing methods [81–83] could be done to characterize metabolic pathways in the HepaRG/HHSteC spheroids, including both glucose producing (e.g. glycolysis and glyconeogenesis) and glucose utilizing (e.g. gluconeogenesis, glycogenolysis) pathways. We may then expand our current computational framework by incorporating more detailed models of hepatic glucose metabolism [50,84] for further elucidation of mechanisms behind T2DM and refined *in silico* predictions. These studies would also provide insight into the extent to which the mechanisms behind insulin resistance in the HepaRG liver-islet MPS are equivalent to those in humans. Future studies should also investigate glucagon dynamics in our HepaRG liver-islet MPS, and its effect on glucose metabolism. However, this would require additional MPS experiments and modifications to the computational model which are outside the scope of this study.

In this study, we used HepaRG/HHSteC spheroids derived from HepaRG cells, a human hepatoma cell line. The HepaRG cell line has been used for almost two decades [85], mostly for drug metabolism and toxicity studies [86–88]. The HepaRG cell line has been shown to exhibit structural and functional features that are representative of primary human hepatocytes (PHHs) for studying glucose and lipid metabolism [89], and disease mechanisms associated to insulin resistance. In particular, HepaRG cells express numerous genes encoding key proteins involved in pathways of glucose and lipid metabolism, including gluconeogenesis, glycogen metabolism, glycolysis, lipid synthesis and degradation [44]. Moreover, a functional insulin responsiveness has been reported for HepaRG cells [44] and shown to be improved by a 3D spheroid co-culture [18]. Therefore, HepaRG cells have been suggested as a suitable model to investigate the regulation of pathways in glucose metabolism and disease mechanisms related to insulin resistance [44,90]. However, the HepaRG cell line does not represent population differences and may display deviations in gene expression from PHHs that are relevant to glucose metabolism, in line with previously reported limitations for their use in drug metabolism and toxicity studies [91]. HepaRG cells have also been shown to exhibit a low expression of the glucagon receptor, and could therefore be unresponsive to glucagon, when cultured in monolayers including only HepaRG cells [89]. However, our 3D spheroid co-culture model might have improved glucagon sensitivity, as previously demonstrated for the improved insulin responsiveness in comparison to monocultures [18]. The use of PHHs, which remain the gold standard for studying hepatic metabolism *in vitro* [92], could improve the physiological relevance of our MPS and the *in vivo* extrapolation of the experimental results, and should be investigated in future studies.

Because of the relatively small number of MPS experiments included in our study, we calibrated the computational model using data exclusively from a given individual experiment. By acquiring data from a larger set of MPS experiments, we could expand our framework with a non-linear mixed-effects (NLME) modelling approach [93,94]. Using this approach, we could estimate the model parameters using data from all the MPS experiments simultaneously, thereby sharing information among them. This would in turn allow us to obtain additional insight on interindividual variability within the experiments, and more robust parameter estimation when the available data from one experiment alone may be insufficient.

The most widespread models to study glucose metabolism under both healthy and T2DM conditions are *in vivo* animal models in rodents. These are comparators for the novel, integrated experimental (MPS)-computational approach presented here. Translation between these animal models and humans is often unsuccessful, partly due to species-specific glucose regulation mechanisms ranging from cell to organ level [12]. To overcome these limitations, the HepaRG liver-islet MPS in this study incorporates human cells at organ emulation levels (e.g. 3D tissue environment), mimicking their human counterpart architecture and function. While our system could potentially be more predictive of *in vivo*, human outcome, its ability to replicate human physiology is still limited. For instance, with our current experimental setting, kinetics of the *in vitro* glucose response was considerably slower than *in vivo*. In our HepaRG liver-islet MPS, glucose levels reached normoglycemia approximately 48 hours after the start of a GTT, as opposed to 1 to 2 hours *in vivo* [66,72,95]. This could be due to several factors including design aspects, properties of the co-culture medium and the organoids, operating conditions (e.g. flow rate) and, most likely to a lesser extent, the lack of other glucose consuming organs (e.g. muscle, brain) and signaling mechanisms (e.g. incretins). For example, the media-to-tissue ratio in our system was in the order of 100:1, whereas the physiological extracellular fluid to tissue ratio is 1:4 [43]. This gives a much larger load of glucose mass per liver cell in the MPS than in the *in vivo* case, likely increasing the time needed to adjust glucose levels back to normoglycemia. The synergistic experimental-computational approach allowed us

to gain insight into the impact of these experimental *in vitro* factors in the system and ultimately bridge the *in vitro-in vivo* gap by compensating for them *in silico*. We argue that to maximize the ability of this approach to exploit the *in vitro* MPS investigations, it should be applied in an iterative fashion that involves two steps: *data interpretation/translation* and *model-guided design*.

In the *data interpretation/translation* step, we calibrated the computational model using glucose and insulin responses measured during GTTs in the HepaRG liver-pancreas MPS and performed model-based extrapolation (i.e. scaling) of *in vitro* experimental aspects (co-culture and organoid volumes, flow rates and incorporation of missing organs) to translate to humans. After this extrapolation, both insulin levels (S3F Fig, shaded areas) and glucose uptake (S3C Fig, shaded areas) were higher than those observed during meal responses in humans (black markers, from Man et al., 2007 [66]). These discrepancies indicate that our current MPS set-up may have design limitations that cannot be compensated for using a purely scaling approach, thereby pinpointing possible biological imperfections related to both the organoids and co-culture conditions. One potential explanation concerning biological imperfections is that the increased insulin levels in the MPS (S3F Fig) are due to impaired hepatic insulin clearance because of the lack of sinusoidal structures in the liver model [96]. Other possible explanations would be enhanced insulin secretion of the pancreatic islets compared to *in vivo*, and the lack of renal insulin clearance mechanisms [96]. As a proof-of-concept investigation, we tested the first hypothesis by comparing the human-translated insulin elimination rate constant in the MPS and the reported value in humans [66], and found it to be approximately four times lower than in humans. We corrected for this discrepancy and proceeded to test the second hypothesis by decreasing the insulin secretion of the pancreatic islets *in silico*. When reducing the insulin secretion rate to half of its initial value, which is within the range of interindividual variability in insulin concentrations found in our study, the simulated glucose and insulin responses agreed well with the experimental data from humans (Fig 8C and 8F). These results suggest that a decreased insulin secretion capacity of the pancreatic islets in the MPS, in combination with increased hepatic insulin elimination rate constant and the model-based scaling approach, would yield human-like responses. We confirmed the validity of the first hypothesis by performing an independent MPS experiment to experimentally measure the value of hepatic insulin clearance *in vitro*. Additional investigations should be performed to further assess the remaining hypotheses in a larger number of experiments and establish a translation on a population level. Similarly, a number of additional biological hypotheses could explain the glucose responses during the GTT initiated at day 13 for the different glycemic regimes (Fig 6A), for example development of insulin resistance due to culture-related factors other than hyperglycemia, or insulin resistance induced by the high-insulin pre-culture conditions.

The mechanistic insights gained from the model could be used in a future *model-guided design* step, to enhance the physiological relevance of our *in vitro* MPS through, for instance, increased insulin clearance by the liver organoid or decreased insulin secretion by the pancreatic islets. This updated MPS set-up would then generate experimental data that supports the development of a respective, extended computational model for prediction in the subsequent *data interpretation/translation* step.

A recent MPS report to advance patient's benefit and animal welfare has identified four elements to make preclinical drug evaluation predictive to human exposure [97]. These elements are: i) academic invention and MPS-model development, ii) tool creation and MPS-model qualification by supplier industries, iii) qualification of a fit-for-purpose assay and its adoption for candidate testing by pharmaceutical industries, and iv) regulatory acceptance of the predictive results of validated assays for a drug candidate for a specific context of use. Here, we propose to support these MPS-based developments with computational modelling. Our results

demonstrate the synergies between MPS and computational models, which we believe would accelerate the drug development cycle. Both in academic science and pharmaceutical decision making, fit-for-purpose experimental-computational models hold the potential to reduce the use of animal models currently used for the same purpose. A purely experimental *in vitro* approach to this goal is further away, since recreating human-like responses *in vitro* poses major challenges related to constraints in design and experimental conditions. For example, the differences in media-to-tissue ratio that lead to the slow time dynamics in GTT responses in our MPS system are difficult to address experimentally, because a reduction in the volume of co-culture medium would result in sampling volumes that are insufficient for analysis. With our integrated experimental-computational approach these limitations can be overcome. Our vision is that once the effect of drugs are well-characterized *in vitro* in MPS recapitulating the physiology of different organs, computational modelling can be used to create *in silico* representations of individual organs that can then be combined in a multi-organ computational model as a step towards extrapolation to humans.

## 5 Conclusion

In conclusion, the use of computational modelling to analyze experimental results from the HepaRG liver-islet MPS allowed for better mechanistic understanding of the physiological processes underlying glucose metabolism and the development of T2DM in the system. Furthermore, it provided a first step towards translation of the experimental responses to *in vivo* outcome, and guidance to improve the physiological relevance of the model. The synergistic experimental-computational approach proposed in this study, when applied in an iterative manner and for a specific context of use, could contribute to eventually generating computational evidence of higher predictive power than that derived from current animal models. We envision that, with further validation, this approach could reduce animal experiments and significantly decrease phase 1 and phase 2 clinical trial failures due to its relatively low cost and ability to generate human-relevant predictions.

## Supporting information

**S1 Fig. Comparison between glucose-stimulated insulin secretion (GSIS) from pancreatic islets exposed to different glycemic levels during the co-culture.** The pancreatic islets were collected from the MPS after 15 days of co-culture. During the co-culture, they were exposed to either hyper- (11mM, red), normo- (5.5 mM, green) or hypoglycemic conditions (2.8 mM, yellow). At day 13, a GTT with a glucose load of 11 mM was performed in all co-cultures. After being collected from the MPS, the pancreatic islets were incubated in low glucose (2.8 mM) over 2h, following 2h incubation in high glucose (16.8 mM). The results correspond to experiments 3 (A), 4 (B) and 5 (C).
(TIF)

**S2 Fig. Agreement between model simulations (lines) and experimental data (markers) for the experiments not shown in the main article (experiments 1 and 4–7). Experiment 1 (A-C):** Glucose concentration in the liver-islet co-culture under hyperglycemia (**A**), glucose concentration in the single-liver culture under hyperglycemia (**B**), insulin concentration in the liver-islet co-culture under hyperglycemia (**C**); **experiment 4 (D-G)**: glucose concentration in the liver-islet co-cultures under normoglycemia (**D**), insulin concentration in the liver-islet co-culture under normoglycemia (**E**), glucose concentration in the liver-islet co-culture under hyperglycemia (**F**), insulin concentration in the liver-islet co-culture under hyperglycemia (**G**); **experiment 5 (H-K)**: glucose concentration in the liver-islet co-culture under normoglycemia

**(H)**, insulin concentration in the liver-islet co-culture under normoglycemia **(I)**, glucose concentration in the liver-islet co-culture under hyperglycemia **(J)**, insulin concentration in the liver-islet co-culture under hyperglycemia **(K)**; **experiment 6 (L-M)**: Glucose concentration in the liver-islet co-culture under normoglycemia **(L)**, insulin concentration in the liver-islet co-culture under normoglycemia **(M)**; **experiment 7 (N-O)**: Glucose concentration in the liver-islet co-culture under hyperglycemia **(N)**, insulin concentration in the liver-islet co-culture under hyperglycemia **(O)**. In hyperglycemic and normoglycemic conditions, co-cultures were exposed to 11 mM or 5.5 mM glucose in each media exchange (arrows), respectively. Model uncertainty is shown as shaded areas in panels A-O. Data in panels A-O are presented as mean ± SEM, where the number of replicas considered for each experiment are: n = 4 (experiments 1, 6 and 7 for all glycemic conditions), n = 5 (experiment 4 for all glycemic conditions and experiment 5 for hyperglycemia) and n = 10 for experiment 5 under normoglycemia.
(TIF)

**S3 Fig. Model predictions of glucose and insulin concentrations in the MPS after scaling to humans, considering the initial secretion rate of the β cells estimated in the co-culture ($\sigma_{max} = 6 \cdot 10^6$ mIU/L/h).** The results correspond to a single experiment (experiment 1). **A,B, D,E**: Model predictions of glucose **(A,B)** and insulin **(D,E)** in the liver and pancreas compartments after scaling. **C** shows the comparison between the model prediction of plasma glucose concentration after scaling and experimental data of glucose response to a meal in healthy subjects [66]. The model-based prediction of the insulin response and the experimental measurements of insulin are compared in **F**. The predictions are computed for the GTT initiated at day 1 (GTT d1-3). The experimental data were acquired in a group of 204 normal subjects [66]. We consider the time point of peak glucose concentration in the experimental data as time = 0 h for this study, since the MPS lacks an intestinal compartment and glucose is administered directly to both the liver and pancreas compartments. Data are presented as mean ± SEM (n = 204). Model uncertainty is depicted as shaded areas in **(A-F)**.
(TIF)

**S4 Fig. Estimation of the hepatic insulin clearance rate in the HepaRG liver-islet MPS based on experimental data from single-islet experiments.** The experimental data corresponds to a single experiment, where single-liver cultures were exposed to hyperglycemic conditions (11 mM glucose in each co-culture medium exchange during a 7-day co-culture period). At days 1 and 6, an insulin dose was added to the co-culture medium to assess insulin clearance. Data in **(A,B)** are presented as mean ± SEM (n = 4). In **(C, D)**, a linear regression model was fitted to the mean values of the experimental data for each day to estimate the hepatic insulin elimination rate constant. The resulting estimated values are $k_{day1}$ = 0.024 (1/h) and $k_{day7}$ = 0.021 (1/h) for days 1 and 7, respectively.
(TIF)

**S5 Fig. Experimental measurements of insulin concentration in an empty chip (i.e. empty culture compartments).** Media exchanges were performed every 24 hours, and samples of the culture media were taken directly after each media exchange (t = 0 h) as well as 24 hours after (t = 24 h). In each media exchange, a specific amount of insulin was added to the culture medium. Data correspond to a single chip replicate (n = 1).
(TIF)

**S1 Table. Summary of the experimental settings used in the MPS experiments and the measurements acquired for calibration and evaluation of the experiment-specific computational models.** GTT: Glucose tolerance test.
(PDF)

**S2 Table. Summary of model evaluations for all the MPS experiments included in the analysis.** Acceptable models could be inferred for every experiment. The threshold for the $\chi^2$ test with 95% significance is calculated based on the number of data points in the experimental time-series for each experiment.
(PDF)

**S3 Table. Estimated values of the experiment-specific parameters for each MPS experiment included in the study.** Values indicated as (-) were not estimated because there were no available data to perform the estimation.
(PDF)

**S1 Appendix. Supplementary methods.** Supplementary description of the computational model, and comparison of hepatic insulin clearance rates in the HepaRG liver-islet MPS with human *in situ* values.
(PDF)

## Author Contributions

**Conceptualization:** Belén Casas, Peter Gennemark, Gunnar Cedersund.

**Data curation:** Belén Casas, Liisa Vilén, Sophie Bauer, Kajsa P. Kanebratt, Charlotte Wennberg Huldt.

**Formal analysis:** Belén Casas, Liisa Vilén, Sophie Bauer, Kajsa P. Kanebratt, Charlotte Wennberg Huldt, Peter Gennemark, Gunnar Cedersund.

**Investigation:** Belén Casas, Liisa Vilén, Sophie Bauer, Kajsa P. Kanebratt, Charlotte Wennberg Huldt, Lisa Magnusson.

**Methodology:** Belén Casas, Liisa Vilén, Sophie Bauer, Kajsa P. Kanebratt, Charlotte Wennberg Huldt, Lisa Magnusson, Uwe Marx, Tommy B. Andersson, Peter Gennemark, Gunnar Cedersund.

**Project administration:** Liisa Vilén, Tommy B. Andersson, Gunnar Cedersund.

**Resources:** Liisa Vilén, Uwe Marx, Tommy B. Andersson.

**Software:** Belén Casas.

**Supervision:** Peter Gennemark, Gunnar Cedersund.

**Validation:** Belén Casas, Liisa Vilén, Sophie Bauer, Kajsa P. Kanebratt, Charlotte Wennberg Huldt, Peter Gennemark, Gunnar Cedersund.

**Visualization:** Belén Casas.

**Writing – original draft:** Belén Casas, Gunnar Cedersund.

**Writing – review & editing:** Belén Casas, Liisa Vilén, Sophie Bauer, Kajsa P. Kanebratt, Charlotte Wennberg Huldt, Lisa Magnusson, Uwe Marx, Tommy B. Andersson, Peter Gennemark, Gunnar Cedersund.

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
