## [Decision Letter · Decision Letter 0]

8 Mar 2022

Dear Dr. Cedersund,

Thank you very much for submitting your manuscript "Integrated experimental-computational analysis of a liver-islet microphysiological system for human-centric diabetes research" for consideration at PLOS Computational Biology.

As with all papers reviewed by the journal, your manuscript was reviewed by members of the editorial board and by several independent reviewers. In light of the reviews (below this email), we would like to invite the resubmission of a significantly-revised version that takes into account the reviewers' comments.

Please pay close attention to Reviewer 2's comments and suggestions when revising the manuscript.

We cannot make any decision about publication until we have seen the revised manuscript and your response to the reviewers' comments. Your revised manuscript is also likely to be sent to reviewers for further evaluation.

Sincerely,

Anders Wallqvist

Associate Editor

PLOS Computational Biology

Jason Papin

Editor-in-Chief

PLOS Computational Biology

Please pay close attention to Reviewer 2's comments and suggestions when revising the manuscript.

Reviewer's Responses to Questions

**Comments to the Authors:**

Reviewer #1: The manuscript

"Integrated experimental-computational analysis of a liver-islet

microphysiological system for human-centric diabetes research" describes an interesting advancement in the modelling and interpretation of microphysiological systems. The specific focus on T2DM is highly relevant and generally the work is well carried out.

The authors should clarify some details to improve the presentation further.

Comments:

In the introduction:

1) Please describe interspecies differences for the specific disease - T2DM - to further motivate the need for MPS in this case

2) The sentence "By integrating 3D cultures of human organ-specific cells in a 87 microfluidic platform, these in vitro systems aim to recreate key microenvironmental aspects of in vivo 88 tissues (flow, multicellular architectures, and tissue-tissue interfaces), thereby being more 89 physiologically relevant than standard cell cultures (9–11). " should be changed as many (most) MPS are not 3D cultures

3) Clearly define what is one exeriment? Is that one system? Where the some of the seven systems run at the same time?

There are some explanations in different parts of the MS, but still the replicates vs experiments are unclear.

4) In section 2.2.1

"

The parameter (mIU/L/h) represents the maximal insulin secretion rate of the sigma_max

cells (i.e. at the beginning of the co-culture). The parameter alpha is estimated based on the experimental

measurements. For large values of this parameter, the decrease in insulin secretion capacity over time

would be negligible."

Describe how you experimentally derive alpha?

5) In fig 4, 6 and 7 - give a metrics for the correlation instead of stating acceptable or good

6) The hypothesis that the pancreatic islets have abnormal insulin secretion due to prolonged exposure to hyperglycemic conditions should be verified with explained experimental conditions

7) In discussion please keep the references format

8) If I understand correctly only HepRG cells were used and not primary hepatocytes - add a discussion how this affect the outcome and in vivo translation of the work

9) The discussion is interesting but a bit too long and general in the end - I suggest that the authors focus on the implications of the specific study

The work would be strengthened by a short conclusion section at the end

Reviewer #2: In the presented paper, Casas and colleagues utilize a microfluidic system with recirculating medium in which two cell compartments containing “liver cells” and human islets are serially connected. This system is then used to derive a mathematical model that can be used to analyze liver-islet crosstalk in vivo. This goal is worthwhile and highly ambitious. Unfortunately however, the manuscript has multiple, at least in part, severe limitations that curb my enthusiasm about this work. Most importantly, as detailed below, the physiological relevance of the model is very limited due to the simplistic fluidic circuit and the use of hepatoma cells instead of primary hepatocytes. Furthermore, the experimental validation is very sparse. For instance, the authors make broad predictions about insulin resistance, islet cell death, insulin secretion rates etc. without having measured any of these parameters, despite the fact that it should be easily possible.

Specific comments:

- The authors state that they “previously presented a two-organ MPS integrating liver and pancreas”, which they now use for further experiments and modeling approaches. However, the authors fail to state that this model uses a hepatic cell line (HepaRG), which cannot serve as a faithful model for the liver as an organ. This fact is not stated in either abstract or introduction and it appears to this reviewer that this is purposefully done in an attempt to mislead the reader regarding the physiological relevance of the utilized model. It is critical that it is made unambiguously clear in title, abstract and introduction that the study uses a hepatoma cell line and not primary liver cells. For instance, “HepaRG” spheroids should be used throughout the manuscript instead of “liver spheroids”.

- In relation to the comment above, IVIVE from cancer cells to human in vivo values seems highly problematic. How can transformed cells be used to study “healthy” conditions?

- Again, related to the overstatement of results and its physiological relevance: the model essentially recirculates medium through a linear connection of HepaRG spheroids and islets, thereby neglecting other tissues that play the predominant role in glucose clearance (skeletal muscle, adipose tissue, CNS). Similarly, there is no/very limited clearance of insulin. Moreover, medium is exchanged every 48h resulting in highly unphysiological, step-wise changes of insulin/glucose levels.

- No information is provided regarding other important pancreatic endocrine signaling molecules, such as glucagon. This should be added.

- The setup raises the question about the added value of such a simple perfusion system. Would there be differences compared to static co-culture of HepaRG spheroids and islets in the same well of e.g. a conventional 384-well plate?

- More information should be provided for the pancreatic islets: Were native islets used or were islets dissociated and reaggregated? Does the model only include endocrine cells or also exocrine cells?

- Scaling factor of liver and islets: If I understand correctly, the authors scale HepaRG cells : islet cells (which should also include alpha and delta cells; possibly also exocrine cells) as 40 : 1. However, the human pancreas has in toto around 1 billion b cells, compared to around 200 billion hepatocytes. Thus, scaling liver:islet seems off by at least a factor of 5. Furthermore, as stated above all the other hundreds of billions of cells in the human body that also take up glucose and insulin are entirely neglected.

- The authors culture their cells in medium containing 860nM insulin. Before transfer, the cells were washed twice and then transferred to insulin-free medium. Given the huge excess of insulin in the pre-culture medium compared to physiological concentrations (around 2nM), more information should be provided regarding the dilution factors of the washed. Furthermore, insulin concentrations should be measured at t0 after transfer to make sure that there was no considerable carry-over and insulin levels are indeed close to zero.

- Model parameters: The rate of beta-cell death was obtained from the literature. However, this is likely highly dependent on the model/platform/culture conditions and should thus be measured in the actual device.

- What is the material of the MPS? Did the authors ever measure insulin absorption/adsorption to the plastic of the culture device. If not, this should be measured in order to not overestimate insulin clearance.

- Promotional company material should not be cited (InSphero, 2016).

- Figure 4A-B: Why is the glucose concentration in the co-culture already at t0 lower than in the monoculture (around 9mM compared to 11mM in the HepaRG-only setup)? This is particularly apparent as the only cycles with measured data (the first and the last) in the co-culture setup start at 9mM, whereas all cycles in between start at 11mM.

- Figure 4E: The authors conclude that “insulin sensitivity in the liver spheroids decreases progressively as they are exposed to hyperglycemic periods during the co-culture (Fig. 4E). This decline in insulin sensitivity leads to reduced glucose utilization by the liver spheroids, resulting in higher daily glucose levels over time.” – However, the model uncertainty does not allow to draw conclusions about increase in insulin resistance and follow-up conclusions are questionable. Rather, insulin sensitivity should be measured directly, e.g. by pAKT Western blotting.

- Figure 5A, 6A: The experimental GTT data (the last cycle) shows no difference between hypo-, normo- and hyperglycemia, thus essentially showing that 2w culture in different glucose levels do not alter insulin sensitivity or that the range of insulin concentrations covered in the device (0.3 to 2.5nM) is not sufficient to alter glucose uptake in HepaRG cells, likely because cells were relatively insulin resistant (due to preculture in 800nM insulin) to begin with.

- Figure 7: The measured values in 7A, B, D and E do not fit the model trends. For instance, there does not seem to be a decrease in insulin concentrations over time (7D-E), whereas there seems to be an increase in glucose levels, which is not predicted by the model (7A-B).

Reviewer #3: It took me a while to understand what the authors were trying to accomplish with this paper, but in the end I was mostly convinced that the work is correct and worthwhile. The goal was to use modeling to establish a microfluidic pseudo-tissue preparation (called MPS) containing human pancreatic islets and liver spheroids and allowing for inter-organ communication as an intermediate platform between in vitro and in vivo approaches. The preparation studied here was previously introduced in Ref. 12 but without modeling. In the present paper, the model of Topp (Ref. #41) for diabetes pathogenesis is applied to the MPS but with significant modification to account for the different glucose-insulin dynamics compared to humans and rodents. The MPS is rich enough to exhibit insulin resistance and impaired insulin secretion in response to hyperglycemia, though these differ from how those phenomena manifest in vivo, as detailed below. I think the MPS is a long way from being usable as a model for type 2 diabetes, but it may be a good first step in that direction.

General Comments:

1. Whereas the Topp model treats insulin resistance as an external forcing factor, implicitly representing consequences of obesity and likely representing resistance mostly in muscle and adipose tissue, the current model looks at insulin resistance in the liver spheroids, attributed to hyperglycemia. It doesn’t seem likely that this plays an important role in human diabetes, but it appears to be involved in the MPS.

2. Insulin-mediated glucose uptake (IMGU) plays a central role in the Topp model, but liver glucose transport does not depend on insulin, Instead, IMGU here represents an increase in diffusion of glucose into the liver as intracellular glucose is converted to glycogen (p. 9). This process is very slow: restoration of basal glucose after a glucose challenge takes 48 hours, in contrast to the 2 hours in vivo (Fig. 4A, text l. 624). This seems to me to result mainly from the lack of muscle tissue in the MPS, and I was surprised that this was not mentioned in the paragraph beginning at l. 615. Please comment.

3. (a) One advance over the Topp model is to make insulin secretion capacity (sigma) time dependent. A priori, it is difficult or impossible to separately identify sigma and beta-cell mass (here, volume) since they appear multiplicatively in Eq. 2, but it works here because the parameters for mass/volume are fixed at the values in Topp’s initial paper (l. 319 - 322, Table 1). Please comment explicitly on this point. I would not call these values “the established range” (l. 452); the Topp model was qualitative and meant to illustrate the key dynamical features, not fit to data.

(b)The model assumes that sigma is a decreasing function of time (Eq. 3), and it does in fact decrease markedly over the 13-day study. However, the formulation precludes the possibility that, like mass, sigma could also show a compensatory increase. Please comment on this modeling choice.

4. Related to point #2: I was impressed but also perplexed by the success of the upscaling to the human in vivo case. The model showed reasonably good agreement, but required doubling of insulin- and non-insulin mediated glucose uptake (to account for the absence of muscle tissue in the MPS), increased insulin clearance (not sure why this was needed) and decreased insulin secretion capacity (no justification was offered for this, other than that it improved the fit). It is further suggested that including endogenous production would have improved the agreement (l. 528).

My enthusiasm was dampened by the number of ad hoc adjustments (though most were plausible), and my perplexity stems from my belief that accounting for muscle is not just a quantitative matter, but involves qualitatively different signaling pathways (insulin-dependent GLUT4). That left me wondering whether the good agreement was fortuitous and whether the envisioned application to drug development for humans is credible. Please comment on this.

Specific Comments:

l. 72 - 73: “over time they are unable to meet the increased insulin demand and T2DM manifests”: It would be better to say “they may be” instead of “they are” because in most cases the beta cells are able to successfully compensate for insulin resistance.

l. 397: “Tab. S3” should be “Tab. S2”.

l.398 - 400: “as opposed to the slower glucose consumption after 13 days”: Please quantify this, as it is difficult to evaluate by eye.

p. 15, top: Please say something about Fig. 4D.

p. 16, description of Fig. 5: why is there no panel for sigma, in contrast to Fig. 4?

p. 17, description of Fig. 6: please acknowledge that even though the predictions of glucose and insulin were validated, there is no obvious way to validate the predictions about insulin resistance, beta-cell volume, or secretion capacity.

l. 511 - 512: the model results in Fig. S3 were described as “in agreement” with those in humans. This is confusing and possibly misleading because the point of Fig. S3 was to say that without the adjustment in secretion capacity, the agreement was poor.

l. 582: “These predictions agree with the previously established hypothesis”: Rather than “agree with” I would say “follow from” so as not to give the impression that the predictions are in agreement with some measurement.

**Have the authors made all data and (if applicable) computational code underlying the findings in their manuscript fully available?**

Reviewer #1: Yes

Reviewer #2: None

Reviewer #3: Yes

PLOS authors have the option to publish the peer review history of their article (what does this mean?). If published, this will include your full peer review and any attached files.

Reviewer #1: No

Reviewer #2: No

Reviewer #3: No
---

## [Decision Letter · Decision Letter 1]

21 Jun 2022

Dear Dr. Cedersund,

Thank you very much for submitting your manuscript "Integrated experimental-computational analysis of a liver-islet microphysiological system for human-centric diabetes research" for consideration at PLOS Computational Biology.

As with all papers reviewed by the journal, your manuscript was reviewed by members of the editorial board and by several independent reviewers. In light of the reviews (below this email), we would like to invite the resubmission of a significantly-revised version that takes into account the reviewers' comments.

We cannot make any decision about publication until we have seen the revised manuscript and your response to the reviewers' comments. Your revised manuscript is also likely to be sent to reviewers for further evaluation.

Sincerely,

Anders Wallqvist

Associate Editor

PLOS Computational Biology

Jason Papin

Editor-in-Chief

PLOS Computational Biology

Reviewer's Responses to Questions

**Comments to the Authors:**

Reviewer #2: The authors have addressed some of my comments very well. However some important concerns remain:

1) While some of the overstatements have been removed, multiple instances that could mislead readers remain. The use of a hepatoma cell line cannot qualify as a model of a liver organ and as such the model is clearly not microphysiological, as it lacks anatomical hepatic structures, other hepatic cells, gradients of oxygen and other important hepatic functions etc. As such, I reiterate that the model should not be called a “liver-islet microphysiological system”. Furthermore, the term “liver spheroids” should be avoided as this term is used in the literature for models of primary human liver cells. Please rephrase to “HepaRG spheroids”.

2) I follow the authors rationale that at least one purpose of the manuscript is to show that a mathematical model might overcome certain limitations of the system that arise from technical issues, such as the lack of models of other glucose sink tissues. This was not clear to me after the first reading and I thank the authors for this clarification. However, the quantification of insulin sensitivity/resistance of HepaRG cells using pAKT Western blot should be conducted as I am concerned that these cells are highly insulin resistant already at the start of the experiment due to long-term culture of cell lines at micromolar insulin concentrations while passaging. However, if those cells do not respond to physiological insulin levels as they should, the system can hardly model in vivo response kinetics.

3) Regarding my previous comment 12: I accept that pipetting errors might indeed accommodate for these differences; however, if all measured values in co-culture are lower than in monoculture, the source of this apparently systematic error should be investigated and not be corrected by simply modeling a different offset.

Reviewer #3: I am satisfied with the responses to the points raised in my first review and I feel the authors have been straightforward in acknowledging the limitations of the preparation as well as its potential as a first step toward improved systems. I endorse in particular the response to Reviewer #2, RC 14: it is reasonable that glucose doesn't rise given the increase in insulin secretion.

One minor comment about the added text on p. 23 on whether beta-cell function rises or falls in response to insulin resistance. It may first rise and then fall, which would be consistent with the observations in animal and human studies that insulin levels first rise and then fall.

**Have the authors made all data and (if applicable) computational code underlying the findings in their manuscript fully available?**

Reviewer #2: None

Reviewer #3: Yes

PLOS authors have the option to publish the peer review history of their article (what does this mean?). If published, this will include your full peer review and any attached files.

Reviewer #2: No

Reviewer #3: No
---

## [Decision Letter · Decision Letter 2]

19 Sep 2022

Dear Dr. Cedersund,

We are pleased to inform you that your manuscript 'Integrated experimental-computational analysis of a HepaRG liver-islet microphysiological system for human-centric diabetes research' has been provisionally accepted for publication in PLOS Computational Biology.

Best regards,

Anders Wallqvist

Academic Editor

PLOS Computational Biology

Jason Papin

Editor-in-Chief

PLOS Computational Biology

Reviewer's Responses to Questions

**Comments to the Authors:**

Reviewer #2: -

**Have the authors made all data and (if applicable) computational code underlying the findings in their manuscript fully available?**

Reviewer #2: None

PLOS authors have the option to publish the peer review history of their article (what does this mean?). If published, this will include your full peer review and any attached files.

Reviewer #2: No

---

## [Editor Report · Acceptance letter]

14 Oct 2022

PCOMPBIOL-D-22-00202R2 

Integrated experimental-computational analysis of a HepaRG liver-islet microphysiological system for human-centric diabetes research

Dear Dr Cedersund,

I am pleased to inform you that your manuscript has been formally accepted for publication in PLOS Computational Biology. Your manuscript is now with our production department and you will be notified of the publication date in due course.

With kind regards,

Agnes Pap
